psychology

observational learning, learning via instructions, evaluative learning, fear conditioning, propositional models

**Author for correspondence:**
Sarah Kasran
e-mail: sarah.kasran@ugent.be

# Learning via instructions about observations: exploring similarities and differences with learning via actual observations

## Sarah Kasran, Sean Hughes and Jan De Houwer

Department of Experimental Clinical and Health Psychology, Ghent University, Gent, Belgium

 SK, 0000-0002-3119-7630; SH, 0000-0001-7689-4272;
JDH, 0000-0003-0488-5224

Our behaviour toward stimuli can be influenced by observing how another person (a model) interacts with those stimuli. We investigated whether mere instructions about a model's interactions with stimuli (i.e. instructions about observations) are sufficient to alter evaluative and fear responses and whether these changes are similar in magnitude to those resulting from actually observing the interactions. In Experiments 1 ($n = 268$) and 2 ($n = 260$), participants either observed or read about a model reacting positively or negatively to stimuli. Evaluations of those stimuli were then assessed via ratings and a personalized implicit association test. In Experiments 3 ($n = 60$) and 4 ($n = 190$), we assessed participants' fear toward stimuli after observing or reading about a model displaying distress in the presence of those stimuli. While the results consistently indicated that instructions about observations induced behavioural changes, they were mixed with regard to whether instructions were as powerful in changing behaviour as observations. We discuss whether learning via observations and via instructions may be mediated by similar or different processes, how they might differ in their suitability for conveying certain types of information, and how their relative effectiveness may depend on the information to be transmitted.

# 1. Introduction

For decades now, learning research has documented the conditions under which regularities in the presence of events influence behaviour (for reviews, see [1,2]). For instance, studies on fear conditioning show that when a neutral conditioned stimulus (CS;

e.g. a light) is paired with an aversive unconditioned stimulus (US; e.g. an electric shock), the CS subsequently elicits a fear response when presented on its own (conditioned response or CR). Similarly, research on evaluative conditioning shows that pairing a neutral CS (e.g. an unknown brand) with a positively or negatively valenced US (e.g. a picture of a cute kitten or a picture of a cockroach) changes people's evaluative responses to the CS (e.g. a brand paired with kittens is liked more than a brand paired with cockroaches; see [3]).

Most learning research has been conducted in non-social contexts and with non-social stimuli. However, research has also looked at how our behaviour changes when we observe a social agent (a *model*) interact with the environment. Starting with seminal work on observational conditioning in rhesus monkeys by Mineka *et al.* [4], a wealth of human and non-human studies now show that the behaviour of a model can function as a 'social' US that influences responses toward a CS that is paired with it. For example, an observer might see that another person behaves fearfully (US) in the presence of a novel animal (CS; e.g. [5]) or shows a painful expression (US) whenever a specific cue (CS) is presented (e.g. [6]). As a result, the CS later evokes a fearful response in the observer (i.e. observational fear conditioning; for reviews, see [7,8]). Threat-related behaviour in a model can also serve to reinstate fearful responses that were previously extinguished via direct experience [9]. Likewise, evaluative responses can be influenced by observing a model's behaviour as well (i.e. observational evaluative conditioning). For instance, people can come to like or dislike a novel person (CS) by simply observing how someone else behaves non-verbally (US) toward that person (e.g. [10–13]; for related findings, see [14,15]). This phenomenon is by no means limited to humans: for example, monkeys can quickly and flexibly come to prefer certain stimuli as a result of observing the behavioural choices of another monkey (for a recent demonstration, see [16]). Put simply, both human and non-human animals can change their behaviour as a result of regularities in the presence of (social) events.

At the same time, humans do not only change their behaviour after observing others but also when they receive from others *verbal information* or *instructions* about the presence of events in the environment. Rather than experiencing that a stimulus is followed by an electric shock, people can simply be told about the stimulus–shock relationship. More than 80 years' worth of research shows that such an instruction can give rise to fear responding (even when no shock is ever administered; see [17] for a recent review). Similarly, instructions about upcoming CS–US pairings can also change how much participants like or dislike the CS (e.g. [18,19]). In addition to learning via direct experience and learning via observation, humans thus have access to a third 'learning pathway' [20], namely learning via instructions.

Surprisingly, as far as we know, all studies on learning via instructions have focused on verbal information about the presence of events in the environment (e.g. 'The name of this cookie will be paired with a positive word'). In principle, however, one could also give verbal information about observations, that is, about how a model interacts with the environment (e.g. 'This person ate this cookie and showed a positive reaction'). Examining the effects of such instructions about observations could not only lead to knowledge about a learning experience that may play an important role in shaping behaviour (i.e. hearing about other people's experiences) but would also allow us to gather more information about how different learning pathways (in this case, learning via observations and learning via instructions) relate to each other in terms of their moderating factors and mediating processes. In developing this line of research, we took inspiration from prior research that compared the effects of receiving verbal information about stimulus pairings with the effects of actually experiencing pairings. In the next section, we therefore highlight some insights provided by that prior research.

## 1.1. Comparing the effects of pairings and instructions about pairings

Although surprising parallels have been found between studies on the effects of actual pairings and studies on the effects of receiving instructions about pairings (see [17] for a review), only a handful of studies directly compared the two within a single study. Most of these studies compared the magnitude of the behavioural changes resulting from these two learning pathways. Some of them suggest that actually being exposed to pairings may have a stronger impact than merely receiving instructions about them (e.g. [21]) and that experiencing pairings after they have already been described has an additive effect on behaviour (e.g. [22,23]). However, other studies suggest that instructions about pairings can be at least as effective, or even more so, than actual pairings (e.g. [19,24,25]). For example, Kurdi & Banaji [19] conducted a series of studies wherein participants either

experienced CS–US pairings, read a description of them, or first read the description and then actually encountered the pairings. They found that instructions led to changes in evaluative responding that were equal or even superior to those resulting from the actual pairings. They also found no benefits of exposure to the pairings if participants had already been informed about them. A similar study in children led to an even more striking result: instructions about pairings had the expected effects, whereas no effects emerged when pairings were presented in the absence of such instructions [26].

These and related studies have attracted much attention because of their theoretical and practical implications. On the practical side, it can be interesting to know whether certain learning pathways are more 'powerful' in shaping behaviour than others (e.g. lead to more intense behavioural responses or responses that are more resistant to change), as this could help to optimize the effectiveness of behavioural interventions. On the theoretical side, the findings can inform cognitive theories of learning, because such theories often include assumptions in terms of the cognitive processes mediating different pathways and in some cases make different predictions regarding the similarity of the resulting effects.

With regard to the latter point, certain single-process associative theories argue that both directly experienced events and instructions can lead to the formation of associations between memory representations of stimuli that (are said to) occur together (e.g. [27–29]). Once a CS–US association has been established, the presentation of the CS not only leads to the activation of its representation, but this activation also spreads to the representation of the US, triggering a CR. It is not clear, however, whether these theories predict differences in the extent to which pairings and instructions result in CS–US associations and thus CRs. In contrast, a single-process propositional perspective assumes that all learning pathways are mediated by the formation and truth evaluation of propositions [30,31]. Propositions differ from associations in a number of ways. First, they are defined in terms of their informational content: they can specify the exact nature of the relation between events. For example, relatively weak fear responding to a blue square could be the result of a participant having formed the proposition that 'the blue square sometimes predicts a mild electric shock', which encodes information about the events themselves (blue square, mild electric shock) as well as about the specific relation between them (sometimes predicts). Second, the holder of a proposition can evaluate the extent to which he or she considers this proposition to be true (i.e. evaluate its truth value). Third, propositions can be used in inferential reasoning (i.e. combined with other propositions to generate new propositions). Finally, the propositional perspective assumes that propositions can be based on a wide range of experiences. Therefore, different pathways can result in the same behaviour change, provided that the content of the information conveyed by those pathways is similar and similar propositions can therefore be formed [32]. Hence, when applied to the comparison of the effects of pairings and instructions about pairings, the propositional perspective predicts that if instructions about pairings convey the same information as actually experienced pairings, the change in behaviour should also be similar.

Other theories assume that multiple processes can drive learning effects. For example, Olsson & Phelps [8] proposed that fear responses can be based on CS–US associations in the amygdala that do not require conscious processing, as well as on associations represented in a distributed network of cortical areas that do involve conscious processing. Whereas the theory assumes that fear conditioning can be mediated by both, learning via instructions is considered to depend exclusively on the latter process and to be subject to certain boundary conditions that do not apply to conditioning (such as conscious awareness of the CS). Some dual-process theories of fear learning also make a strong distinction with regard to the type of outcome of different processes, assuming that amygdala-based associations produce automatic responses (such as physiological responses), whereas self-reports of fear would be produced by different mechanisms (e.g. [33]). The theory of Olsson and Phelps does not make such a strong distinction. Instead, it assumes that cortical associations created by verbal instructions can also influence physiological responses, be it in a way that is more indirect and therefore less powerful than the impact of amygdala-based associations that are created by repeated pairings. Taken together, dual-process theories of fear responding would seem to predict that instructions should have weaker effects on automatic fear responses than actual pairings.

Similarly, to explain evaluative responses, dual-process theories often distinguish between associative and propositional mechanisms, assuming that these vary in their sensitivity to specific experiences (pairings versus instructions) and their influence on evaluative responses measured under conditions suboptimal for cognitive processing (often referred to as automatic evaluations). Again, some theories make a strong distinction. For example, Strack & Deutsch [34] proposed that instructions can influence evaluations only via a propositional system that requires sufficient cognitive capacity, thus predicting

that instructions should not influence automatic evaluations. Other theories merely assume that automatic evaluations are less sensitive but not impervious to verbal instructions [35] or that automatic evaluations are only indirectly influenced by instructions via the influence of the propositional process on the associative process [36]. Given these assumptions, dual-process theories of evaluative learning would predict instructions to generally have weaker effects on automatic evaluations than pairings.

In sum, although not all available theories can be used to derive straightforward predictions, empirical knowledge about similarities and differences between the effects of actual pairings and instructions about pairings can inform and constrain theorizing about the cognitive processes that mediate these two learning pathways.

## 1.2. Comparing the effects of observations and instructions about observations

Against this backdrop of research comparing the effects of actual pairings and instructions about pairings, we examined for the first time what happens when instructions about *observed* events are provided. That is, we informed participants about regularities between stimuli and a model's behaviour as they would be encountered during an observational conditioning procedure. If such instructions about observations induce clear behavioural changes, this would be an interesting finding as such because it would demonstrate the impact of a highly indirect learning experience. That is, instructions about observations can be seen as one indirect learning pathway (i.e. instructions) providing information about another indirect learning pathway (i.e. observations). Demonstrating their effects would therefore further highlight the remarkable capacity of humans to learn in an indirect manner (i.e. in the absence of any direct personal experiences).

Additional important information can be gained from directly comparing these effects to the effects of actual observations. Similar to how studies which compared learning via pairings and learning via instructions about pairings can inform theories that include assumptions about the processes mediating these two types of learning (see above), comparing learning via observations and learning via instructions about observations may inform theories that include assumptions about how the mechanisms driving observational learning relate to those driving instructed learning. Although not all of the theoretical perspectives discussed earlier include explicit assumptions about the observational learning pathway, some do. Single-process perspectives assume that all pathways, including learning via observations, are mediated by the same process (although only the propositional perspective clearly predicts that their effects should therefore be similar as long as the information is the same). In contrast, the theory of Olsson & Phelps [8] assumes that the processes mediating observational fear conditioning partially diverge from those mediating instructed fear learning and are instead highly similar to the processes involved in direct fear conditioning (with the exception that social cognition mechanisms are likely to be involved in observational learning). Finally, a related distinction that has recently been highlighted in this context is the computational distinction between model-based and model-free learning. The former involves an internal model of the environment[1] that can be updated instantly (similar to how the propositional perspective assumes that propositions about relations between events can be changed based on a single instruction); the latter does not involve such a model and thus requires (additional) learning trials in order for behaviour to change. Whereas learning via observation has been assumed to depend mostly on model-free learning (especially in simple cases) and only in some instances on model-based learning, learning via instructions would always seem to require model-based learning [37].

In sum, once again the available theories seem to be divided into single-process and dual-process perspectives. Comparing the effects of instructions and observations may inform these theories. For example, although most dual-process theories would likely still be able to explain these two pathways having highly similar effects, such a finding would be more in line with a single-process perspective. While there have been a few studies that included a comparison between observations and instructions [6,25], the informational content always differed between the two pathways (i.e. the instructions were about future direct experiences rather than about a model's reactions to stimuli), meaning that any discrepancies in their effects could have been due to the difference in *content* rather than due to different *processes* being at play. In contrast, we conducted a more direct comparison because the instructions described the same information as one would encounter during an observation phase. This also more closely parallels the earlier comparison between directly

---

[1]Not to be confused with the 'model' in the sense of the social agent observed during observational learning.

experienced pairings and instructions about those pairings, where the instructions are generally designed to convey the information encountered in the direct experience condition. In sum, by documenting the similarities and differences between learning via observations and learning via instructions about observations and exploring potential moderators of those differences, the outcomes of the current direct comparison may be able to inform theorizing about the similarity of the underlying processes.

## 1.3. The current research

Across four experiments, we tested whether instructions about observations would lead to behavioural changes in terms of evaluative (Experiments 1 and 2) and fear responses (Experiments 3 and 4). As the introduction illustrated, theoretical perspectives mostly differ in their predictions regarding the effect of instructions on responses that are more automatic in nature. Therefore, we included not only self-reports but also a measure of automatic evaluations in Experiments 1 and 2 (a personalized implicit association test (pIAT); [38]) and a physiological index of fear in Experiment 3 (skin conductance responses (SCRs)).

In addition, we compared the effects of instructions about observations with the effects of actually observing the events. This allowed us to look at similarities and differences between the two pathways as well as potential moderators of those differences, which could have both theoretical and practical implications. To guide our research, we explored predictions of a propositional perspective. Specifically, to the extent that actually observed events and instructions about those observed events can be expected to result in the same or a similar proposition being formed and considered valid, this perspective predicts that both pathways would have a similar impact, regardless of whether self-reports or more automatic responses are measured.[2]

For all experiments, we registered our hypotheses, planned sample size, procedural details and planned analyses on the Open Science Framework (Experiment 1: https://osf.io/hgcfy/; Experiment 2: https://osf.io/9v3cm/; Experiment 3: https://osf.io/7yt9z/; Experiment 4: https://osf.io/uw3q2/). These registrations were followed unless otherwise specified. Materials, raw data, processed data and all R code used for data processing and analysis are available on the OSF page (https://osf.io/ay25z).

# 2. Experiment 1

Experiment 1 focused on evaluative responses and was designed to examine three questions. First, do instructions about observations lead to changes in evaluative responding? Second, are these changes in evaluative responding as large as those that result from actual observations? Third, given that CS–US pairings have previously been found to have no additive effect when participants had already been informed about these pairings [19], is there an added value of actually observing the regularities after receiving instructions about them?

Participants were divided into three groups. In the 'observations' condition, participants watched videos of a model who tasted two cookies (referred to as 'Empeya' and 'Plogo') and reacted positively to one cookie (the positive CS; $CS_{pos}$) and negatively to the other (the negative CS; $CS_{neg}$). We then assessed their evaluations of both cookies by asking them (i) to provide ratings of how much they expected to like each cookie and (ii) to complete a pIAT [38], a task that assesses automatic evaluations based on the speed (reaction time) with which participants can categorize target stimuli (in this case, the names of the two cookies) using the same keys as liked or disliked words.

The observations group was compared with two other groups. In the 'instructions' condition, participants were told that they would later watch videos in which a person tasted cookies and were informed which reaction the person would show to each cookie. Critically, however, they never watched those videos. Finally, in the 'combined' condition, participants first received the aforementioned instructions and then watched the videos. Based on a propositional perspective, we

---

[2]As may have become clear from our earlier discussion, the propositional perspective is quite broad and not formalized (hence the term 'perspective'). As a result, it is difficult (if not impossible) to derive predictions that would allow one to unequivocally falsify this perspective (see also [32,39]). We do not consider this to be a problem for the current research because our goal was not to falsify any given theoretical perspective (in fact, as one reviewer pointed out, there are strong similarities between the information assumed to be encoded in propositions and the information assumed to be contained in the internal model of the environment in model-based reinforcement learning (e.g. [37]). Rather, we set out to document an interesting phenomenon and to generate empirical knowledge about this phenomenon, which could have practical implications as well as inform future theoretical thinking (in the sense that relevant theories would need to take the generated findings into account). The propositional perspective simply served as a guide to give the current research direction.

predicted that all three conditions would have a similar impact on evaluations of the cookies (with the $CS_{pos}$ being evaluated more positively than the $CS_{neg}$).

## 2.1. Method

### 2.1.1. Participants and design

Based on a power analysis indicating that a total sample size of $n = 258$ was required in order to have 80% power to detect medium-sized differences in follow-up comparisons (with $\alpha = 0.016$ to correct for multiple comparisons), and that we estimated around 10% exclusions based on prespecified pIAT performance criteria, we planned to recruit a sample of $n = 288$ participants via Prolific Academic (https://www.prolific.co/). Slightly more participants completed the experiment due to a server lag, resulting in complete data for $n = 292$ participants (177 men, 113 women, 2 non-binary people; $M_{age} = 28.70$, s.d.$_{age} = 7.96$).

We used a between-subjects design with three levels for acquisition type: observations, instructions and combined. Stimulus assignment (whether Empeya or Plogo served as the $CS_{pos}$), task order (whether participants first completed the ratings or the pIAT) and pIAT block order (whether participants first completed the learning-consistent or the learning-inconsistent block of the pIAT) were counterbalanced across participants.

### 2.1.2. Materials

#### 2.1.2.1. Videos

Two source videos were used, one showing a positive reaction and one showing a negative reaction. In both videos, the model (a 23-year-old man) took a cookie from a plate, took a bite and displayed a positive or negative reaction for approximately 5 s. A label placed next to the plate clearly showed the name of the cookie (i.e. Empeya or Plogo). These two source videos were selected from a larger set of videos based on pre-ratings in terms of valence and believability that were obtained in preparation for another study (pre-rating materials and data are available at https://osf.io/4vbxz/). The two source videos were edited to vary the name on the label in order to counterbalance stimulus assignment.

#### 2.1.2.2. Personalized implicit association test

The target stimuli used in the pIAT consisted of six versions of each CS name (in lower- or upper-case and regular, bold or italic font) presented in Arial. The two CS names (Empeya and Plogo) served as labels for classifying these stimuli into the two categories. The attribute stimuli were six positive (*Pleasure, Holidays, Rainbows, Gifts, Peace* and *Friends*) and six negative (*Sickness, Accidents, Abuse, Death, Fear* and *Pain*) words in Arial Black. The words 'I like' and 'I dislike' served as labels for classifying these attribute stimuli.

### 2.1.3. Procedure

The experiment was programmed in Inquisit 4.0 and hosted via Inquisit Web (Millisecond Software, Seattle, WA). After providing demographic information, all participants were told that we were working with a start-up company that produced two new cookies (referred to as Empeya and Plogo) and that we had recorded videos of a person who was asked to eat these cookies and to clearly display whether he liked or disliked them. Participants then proceeded to the acquisition phase, completed the evaluative measures and answered a number of exploratory questions.

#### 2.1.3.1. Acquisition phase

The acquisition phase depended on the between-subjects manipulation (figure 1 shows an overview of the different acquisition types). In the *observations condition*, participants watched one video in which the model reacted positively to the $CS_{pos}$ by showing a facial expression of enjoyment and taking a second bite, and a second video in which the model reacted negatively to the $CS_{neg}$ by displaying disgust via his facial expression and body language. Both videos were presented three times in a random order with an inter-trial-interval (ITI) of 3 s.

In the *instructions condition*, participants were told that later on in the experiment, they would watch two videos several times. They were then given a description of the regularities, depending on the counterbalanced stimulus assignment. For example, participants for whom Empeya served as the $CS_{pos}$

(*a*)

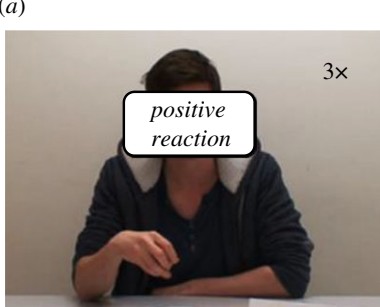 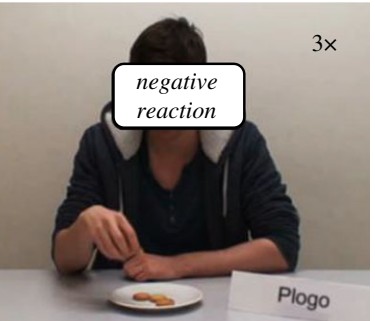

(*b*)

In one video, the person eats the EMPEYA cookie and shows a POSITIVE reaction.

In the other video, the person eats the PLOGO cookie and shows a NEGATIVE reaction.

(*c*)

In one video, the person eats the EMPEYA cookie and shows a POSITIVE reaction.

In the other video, the person eats the PLOGO cookie and shows a NEGATIVE reaction.

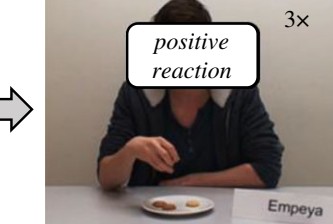 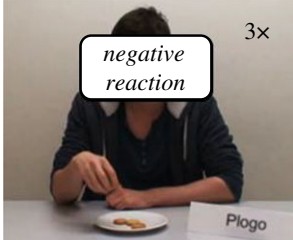

(*d*)

This is what the person's positive reaction looks like:

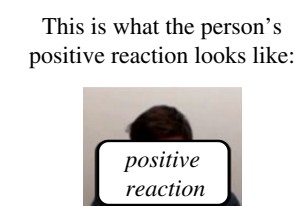

This is what the person's negative reaction looks like:

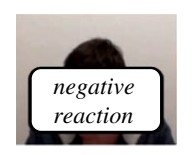

In one video, the person eats the EMPEYA cookie and shows a POSITIVE reaction.

In the other video, the person eats the PLOGO cookie and shows a NEGATIVE reaction.

**Figure 1.** Overview of the acquisition types in Experiments 1 and 2. (*a*) Observations condition (Experiments 1 and 2), (*b*) instructions condition (Experiments 1 and 2), (*c*) combined condition (Experiment 1) and (*d*) enhanced-instructions condition (Experiment 2). Because we do not have consent from the actor to publish images from the videos in their original form, the actor's face has been masked with labels in this figure. Naturally, this was not the case in the videos shown to participants. In the examples provided in this figure, Empeya served as the $CS_{pos}$ while Plogo served as the $CS_{neg}$. As stimulus assignment was counterbalanced, half of the participants encountered this combination while the other half encountered the opposite combination (i.e. Plogo as the $CS_{pos}$ and Empeya as the $CS_{neg}$).

read the following instructions: 'In one video, the person eats the EMPEYA cookie and shows a POSITIVE reaction. In the other video, the person eats the PLOGO cookie and shows a NEGATIVE reaction'. They were then told that they would complete a number of tasks before they would watch these videos.

In the *combined condition*, participants were told that in a minute, they would watch two videos several times. They then read the same description of the regularities as the instructions group. Unlike the instructions group, however, they watched the videos immediately after (i.e. prior to completing the evaluative measures).

### 2.1.3.2. Self-reports

Eight questions (four per CS) were presented in a random order. Participants were asked to indicate on scales from −10 to +10 what they thought their opinion of the CS would be (from *very bad* to *very good*, and from *very negative* to *very positive*), how much they thought they would like the CS (from *I would dislike it very much* to *I would like it very much*), and how pleasant or unpleasant they thought they would consider the CS to be (from *very unpleasant* to *very pleasant*). Zero was indicated as a neutral midpoint.

### 2.1.3.3. Personalized implicit association test

The pIAT consisted of 180 trials. On each trial, a stimulus was presented in the middle of the screen and participants had to use the D and K keys on their keyboard to classify this stimulus as quickly as possible according to labels at the top left (D) and top right (K) of their screen. On target trials, participants had to classify the names 'Empeya' and 'Plogo' into their respective categories; on attribute trials, participants had to classify positive and negative words in terms of whether they liked or disliked them. On target trials, incorrect responses were followed by error feedback (a red 'X' presented for 200 ms) before the trial ended (ITI: 400 ms).

The pIAT was divided into seven blocks. Before each block, participants were informed of the response mappings for that block and reminded to respond as quickly and accurately as possible. Block 1 consisted of 20 target trials: participants had to sort Empeya and Plogo into their respective categories. Block 2 consisted of 20 attribute trials: participants had to sort valenced words in terms of whether they liked or disliked them. These initial blocks allowed participants to practise the response mappings for both trial types. Block 3 (20 trials) combined the two trial types: on some trials, participants had to sort the CSs into the CS categories and on other trials, they had to sort words in terms of whether they liked or disliked them. Block 4 consisted of 40 trials but otherwise had the same structure as Block 3. In Block 5, participants again practised sorting the CS names; however, the response mapping for the CS categories was now reversed relative to the previous blocks. Block 6 (20 trials) again combined the two trial types but with the 'new' response mapping for target trials. Finally, Block 7 consisted of 40 trials but otherwise had the same structure as Block 6. Trial order within each block was random and the relevant labels remained on top of the screen throughout each block.

Because pIAT block order was counterbalanced, for half of the participants, the initial response mappings were *consistent* with the acquisition phase (i.e. sorting the $CS_{pos}$ with the same key as liked words and sorting the $CS_{neg}$ with the same key as disliked words) whereas for the other half, the initial response mappings were *inconsistent* with the acquisition phase (i.e. sorting the $CS_{pos}$ with the same key as disliked words and sorting the $CS_{neg}$ with the same key as liked words).

### 2.1.3.4. Exploratory questions

Before this final phase, participants in the instructions group were informed that they would not actually watch the videos. Depending on the condition they were in, participants were then asked whether they had read the instructions (instructions and combined conditions), as well as which reaction the model had shown to each CS or had been said to show in the instructions. They were also asked to indicate on scales from 0 to 10 how believable they considered the videos to be (observations and combined conditions), to what extent they had believed they would watch the videos (instructions condition), and how much they thought their ratings and their pIAT performance had been influenced by the videos and/or by the instructions. Finally, they were asked to type in what they believed our hypothesis to be and to indicate whether their behaviour on the self-reports and on the pIAT had been driven by demand compliance or reactance (the response options being 'Yes', 'No' and 'I don't know'). Participants were then fully debriefed and thanked for their participation.

## 2.2. Results

### 2.2.1. Data preparation

We first excluded the data of participants who provided incomplete data or who reported technical issues during their participation ($n = 11$), as well as one participant who was not fluent in English. This resulted in complete data for 292 participants (see Participants and design). We then excluded participants who reported not reading the instructions ($n = 3$), who made more than 30% errors across the entire pIAT ($n = 4$), who made more than 40% errors on any of the combined blocks ($n = 16$) or

who completed more than 10% of pIAT trials faster than 300 ms ($n = 1$). The final sample consisted of 268 participants (160 men, 106 women, 2 non-binary people; $M_{age} = 28.72$, s.d.$_{age} = 7.82$).

The evaluative ratings were averaged to create two mean scores, one for the $CS_{pos}$ and one for the $CS_{neg}$. We then subtracted the $CS_{neg}$ score from the $CS_{pos}$ score to create a mean difference score (i.e. a larger difference score indicated a stronger preference for the $CS_{pos}$ over the $CS_{neg}$). Reaction times on the pIAT were used to calculate participant-level scores according to the D1-algorithm [40], such that positive pIAT scores reflected a more positive evaluation of the $CS_{pos}$ relative to the $CS_{neg}$ whereas negative scores reflected the opposite.

### 2.2.2. Data analysis

#### 2.2.2.1. Analytic strategy

To assess the impact of each manipulation on evaluative responding, we conducted one-sided, one-sample $t$-tests to examine whether the rating difference scores and pIAT scores were larger than zero in each condition (i.e. if the $CS_{pos}$ was evaluated more positively than the $CS_{neg}$). To compare the different conditions, we ran an analysis of variance (ANOVA) on each dependent variable to test for a main effect of acquisition type. If the ANOVA indicated that scores indeed differed as a function of acquisition type, we then conducted pairwise comparisons to test which groups differed from one another (using Holm–Bonferroni correction). Finally, we used the Akaike information criterion (AIC) to assess if any of the counterbalanced factors improved model fit, and if so, we tested whether the effect of acquisition type remained significant in an ANOVA that included these factors (see the electronic supplementary material for full models and results). All hypothesis tests were conducted at the $\alpha = 0.05$ significance level. Ninety-five per cent confidence intervals are reported for Cohen's $d$ and 90% confidence intervals are reported for $\eta_p^2$. Finally, we also report Bayes factors ($BF_{10}$) which represent the probability of the alternative hypothesis compared with the null hypothesis given the observed data [41].

#### 2.2.2.2. Main analyses

Participants in all three conditions rated the $CS_{pos}$ more positively than the $CS_{neg}$: the difference score was significantly larger than zero in the observations group ($M = 9.89$, s.d. $= 7.80$), $t_{95} = 12.43$, $p < 0.001$, $d = 1.27$, [1.00, 1.54], $BF_{10} > 10\,000$, the instructions group ($M = 6.13$, s.d. $= 7.99$), $t_{85} = 7.11$, $p < 0.001$, $d = 0.77$, [0.52, 1.01], $BF_{10} > 10\,000$ and the combined group ($M = 9.62$, s.d. $= 5.94$), $t_{85} = 15.01$, $p < 0.001$, $d = 1.62$, [1.29, 1.94], $BF_{10} > 10\,000$. Participants' pIAT performance also indicated a more positive evaluation of the $CS_{pos}$ relative to the $CS_{neg}$: pIAT scores were larger than zero in the observations group ($M = 0.34$, s.d. $= 0.42$), $t_{95} = 7.98$, $p < 0.001$, $d = 0.81$, [0.58, 1.04], $BF_{10} > 10\,000$, the instructions group ($M = 0.17$, s.d. $= 0.42$), $t_{85} = 3.76$, $p < 0.001$, $d = 0.41$, [0.18, 0.62], $BF_{10} = 135.6$ and the combined group ($M = 0.35$, s.d. $= 0.43$), $t_{85} = 7.54$, $p < 0.001$, $d = 0.81$, [0.57, 1.06], $BF_{10} > 10\,000$. In sum, all three conditions led to the expected changes in self-reported and automatic evaluations.

The size of the difference between the ratings of the $CS_{pos}$ and the $CS_{neg}$ varied as a function of acquisition type, $F_{2,265} = 7.23$, $p < 0.001$, $\eta_p^2 = 0.05$, [0.01, 0.10], $BF_{10} = 25.11$ (figure 2a). Specifically, the difference was significantly smaller in the instructions condition than in the observations condition, (corrected) $p = 0.002$, $BF_{10} = 17.93$, and the combined condition, $p = 0.004$, $BF_{10} = 20.26$. Scores in the observations and combined conditions did not differ from each other, $p = 0.81$, $BF_{10} = 0.17$. The main effect of acquisition type remained significant when the model was updated based on AIC values, $F_{2,260} = 7.19$, $p < 0.001$, $BF_{10} = 26.87$.[3]

pIAT scores also differed as a function of acquisition type, $F_{2,265} = 5.06$, $p = 0.007$, $\eta_p^2 = 0.037$, [0.006, 0.076], $BF_{10} = 3.69$ (figure 2b). Similar to the pattern found for self-reports, the instructions group showed a smaller effect than both the observations group, $p = 0.017$, $BF_{10} = 5.33$, and the combined group, $p = 0.017$, $BF_{10} = 5.74$, while the latter two groups did not differ from each other, $p = 0.89$, $BF_{10} = 0.16$. The main effect of acquisition type remained significant when the model was updated based on AIC values, $F_{2,262} = 5.34$, $p = 0.005$, $BF_{10} = 4.57$.[4]

---

[3]For completeness, please note that this main effect was qualified by an interaction with stimulus assignment, $F_{2,260} = 3.51$, $p = 0.03$, $BF_{10} = 1.34$, such that the effect of acquisition type was significant only if Plogo served as the $CS_{pos}$. However, this interaction was likely spurious given the small BF.

[4]For completeness, please note that this main effect was qualified by an interaction with block order, $F_{2,262} = 3.31$, $p = 0.038$, $BF_{10} = 1.23$, such that it was significant only if participants completed the learning-consistent block first. However, this is again probably a spurious finding given the small BF.

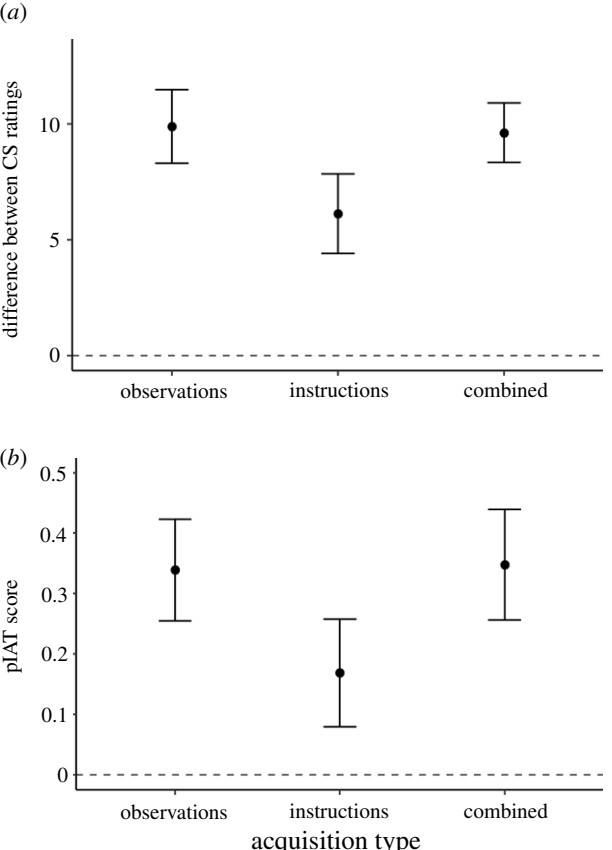

**Figure 2.** (*a*) Difference between $CS_{pos}$ and $CS_{neg}$ ratings as a function of acquisition type (Experiment 1). (*b*) pIAT scores as a function of acquisition type (Experiment 1).

### 2.2.2.3. Exploratory analyses

Some points regarding participants' answers to the exploratory questions are worth mentioning (see the electronic supplementary material for all exploratory analyses). First, most participants correctly remembered the (observed or instructed) pairings, and the results of our main analyses were unchanged when we included only participants with perfect memory for the pairings ($n = 246$). Second, most participants in the instructions group indicated that they had believed they would watch the videos ($M = 8.26$). Therefore, the smaller effects in the instructions group are unlikely to be due to this group not remembering the instructions or not believing that they would actually watch the videos. Finally, the rating results were unchanged when we included only participants who reported that their ratings had not been influenced by demand compliance ($n = 186$) or only participants who reported that their ratings had not been influenced by reactance ($n = 174$). Therefore, it is unlikely that the positive rating of the $CS_{pos}$ relative to the $CS_{neg}$ was simply an artefact of trying to comply with or resist the perceived experimenter demand.

## 2.3. Discussion

In Experiment 1, we examined if instructions about observations would have an impact on likes and dislikes, whether the magnitude of the impact would be comparable to that of actual observations, and if there would be any added benefit of combining observations with instructions. We found that all three manipulations induced changes in liking as reflected by evaluative ratings and pIAT scores. Although instructions were enough to change what people liked or disliked, these effects were significantly smaller than the effects of observations or observations combined with instructions. Thus, it appears that actually observing regularities between stimuli and a model's reactions influences evaluations to a greater extent than simply being told about those regularities.

One potential reason for this weaker effect of instructions relative to observations is that our instructions might have been somewhat vague. We presented two sentences that simply stated that the model would show a 'positive' reaction to the $CS_{pos}$ and a 'negative' reaction to the $CS_{neg}$.

Without any further information, participants have no way of knowing *how* positive or negative the model's reactions are (especially in the case of cookies they might assume the negative reaction to be a rather subtle one, while the reaction in the actual videos was fairly strong). Therefore, because the instructions and the observations conditions differed not only in terms of the format in which information was presented, but also in terms of the quality of that information, the obtained difference between the conditions may not reflect the impact of the pathways themselves but rather of the information that was conveyed.

Therefore, in Experiment 2, we examined if the difference between observations and instructions would remain when participants in the instructions condition were given the opportunity to see what a 'positive' or 'negative' reaction actually referred to, before they were informed about the regularities between those reactions and the two cookies. This also more closely resembles research that compared evaluative conditioning and instructed evaluative conditioning, in which participants are often shown all of the CSs and USs before receiving instructions about how they will be paired (e.g. [19]).

# 3. Experiment 2

We again exposed participants to the same observations and instructions conditions as in Experiment 1. However, we now added an 'enhanced-instructions' condition. In this condition, participants first saw cropped videos of the model's positive or negative facial expressions, allowing them to gain a sense of the nature and intensity of the model's reactions. Afterwards they received the same instructions as the instructions group, indicating that the $CS_{pos}$ would be followed by the positive reaction and the $CS_{neg}$ would be followed by the negative reaction. Based on a propositional perspective, we would predict that the effects in this enhanced-instructions condition would be (i) larger than the effects in the instructions condition and (ii) similar in magnitude to the effects in the observations condition (given that the information they conveyed was now closer in nature).

## 3.1. Method

### 3.1.1. Participants and design

We recruited participants on Prolific until we had complete data for 288 participants (157 men, 130 women, 1 non-binary person; $M_{age} = 28.33$, s.d.$_{age} = 7.59$). The design was similar to Experiment 1, except that participants were assigned to an observations condition, an instructions condition or an 'enhanced-instructions' condition (i.e. the combined condition of Experiment 1 was replaced by this new condition).

### 3.1.2. Materials

#### 3.1.2.1. Videos
The videos shown to the observations group were identical to those in Experiment 1. In addition, edited versions of these videos were created for the enhanced-instructions condition by (i) cutting the videos so that they started only after the model had taken his first bite of the cookie and (ii) cropping them so that they only showed the model's face rather than the entire setting. Consequently, the edited videos showed only the positive or negative reaction, not the name of the corresponding CS.

#### 3.1.2.2. Personalized implicit association test
The pIAT was identical to the task used in Experiment 1.

### 3.1.3. Procedure

Overall, the procedure was similar to that of Experiment 1, with one key difference. Participants in the observations conditions and instructions conditions completed the same manipulations as their counterparts in Experiment 1. However, the combined condition was replaced by the enhanced-instructions condition (figure 1). Participants in this condition were informed that they would soon see videos in which the model reacted positively to one cookie and negatively to another (similar to the instructions group). However, unlike the instructions group, they were first shown cropped videos

of the model emitting the positive and the negative reactions and only then read the instructions about the regularities between the CSs and the model's reactions.

## 3.2. Results

### 3.2.1. Data preparation

We first excluded the data of participants who provided incomplete data or who reported technical issues ($n = 13$). This resulted in complete data for 288 participants (see Participants and design). We then excluded participants who had completed some parts of the experiment twice ($n = 2$), who reported not having read the instructions ($n = 2$), who made more than 30% errors across the entire pIAT ($n = 5$), who made more than 40% errors on any of the combined blocks ($n = 18$) or who completed more than 10% of pIAT trials faster than 300 ms ($n = 1$). The final sample consisted of 260 participants (140 men, 119 women, 1 non-binary person; $M_{age} = 28.45$, s.d.$_{age} = 7.70$). Difference scores and pIAT scores were calculated in the same way as in Experiment 1.

### 3.2.2. Data analysis

The analytic strategy was identical to that of Experiment 1.

#### 3.2.2.1. Main analyses

The $CS_{pos}$ was rated more positively than the $CS_{neg}$ in all three conditions: the difference between the CS ratings was larger than zero in the observations group ($M = 10.59$, s.d. $= 6.63$), $t_{82} = 14.56$, $p < 0.001$, $d = 1.60$, [1.27, 1.92], $BF_{10} > 10\,000$, the instructions group ($M = 5.34$, s.d. $= 6.78$), $t_{88} = 7.43$, $p < 0.001$, $d = 0.79$, [0.55, 1.02], $BF_{10} > 10\,000$ and the enhanced-instructions group ($M = 7.91$, s.d. $= 7.26$), $t_{87} = 10.23$, $p < 0.001$, $d = 1.09$, [0.82, 1.35], $BF_{10} > 10\,000$. The same was true for pIAT scores (observations group: $M = 0.31$, s.d. $= 0.40$, $t_{82} = 7.09$, $p < 0.001$, $d = 0.78$, [0.53, 1.02], $BF_{10} > 10\,000$; instructions group: $M = 0.28$, s.d. $= 0.43$, $t_{88} = 6.11$, $p < 0.001$, $d = 0.65$, [0.42, 0.87], $BF_{10} > 10\,000$; enhanced-instructions group: $M = 0.29$, s.d. $= 0.43$, $t_{87} = 6.31$, $p < 0.001$, $d = 0.67$, [0.44, 0.90], $BF_{10} > 10\,000$). In sum, all manipulations had the expected impact on self-reported and automatic evaluations.

The ratings varied as a function of acquisition type, $F_{2,257} = 12.44$, $p < 0.001$, $\eta_p^2 = 0.09$, [0.04, 0.14], $BF_{10} = 2329.66$ (figure 3a). Specifically, all three conditions differed significantly from each other. Replicating the pattern of Experiment 1, the effect in the observations condition was larger than the effect in the instructions condition, $p < 0.001$, $BF_{10} > 10\,000$. Although the effect in the enhanced-instructions group was slightly larger than that in the instructions group, $p = 0.024$, $BF_{10} = 2.51$, it was still not as large as that in the observations group, $p = 0.024$, $BF_{10} = 2.99$. In sum, the self-reported effect was largest in the observations condition, slightly smaller in the enhanced-instructions condition, and still smaller in the instructions condition. The main effect of acquisition type remained significant when the model was updated based on AIC values, $F_{2,256} = 13.91$, $p < 0.001$, $BF_{10} = 8200$.

In contrast, pIAT scores did not differ as a function of acquisition type, $F_{2,257} = 0.16$, $p = 0.86$, $\eta_p^2 = 0.001$, [0.00, 0.01], $BF_{10} = 0.05$ (figure 3b). All Bayes factors for the follow-up comparisons favoured the null hypothesis (observations versus instructions: $BF_{10} = 0.19$; observations versus enhanced-instructions: $BF_{10} = 0.18$; instructions versus enhanced-instructions: $BF_{10} = 0.16$). In other words, there was evidence for the *absence* of differences between groups, suggesting that all three conditions led to similar pIAT effects. In addition, the main effect of acquisition type was no longer included when the model was updated based on AIC values.

#### 3.2.2.2. Exploratory analyses

Two points are noteworthy (see the electronic supplementary material for all exploratory analyses). First, our main results were again unchanged when we included only participants who had perfect memory for the regularities ($n = 221$). Second, the pairwise comparison between the instructions and the enhanced-instructions conditions became non-significant when we only included participants who reported no demand compliance ($n = 163$). When we only included participants who reported no reactance ($n = 178$), only the difference between the observations and the instructions conditions remained significant.

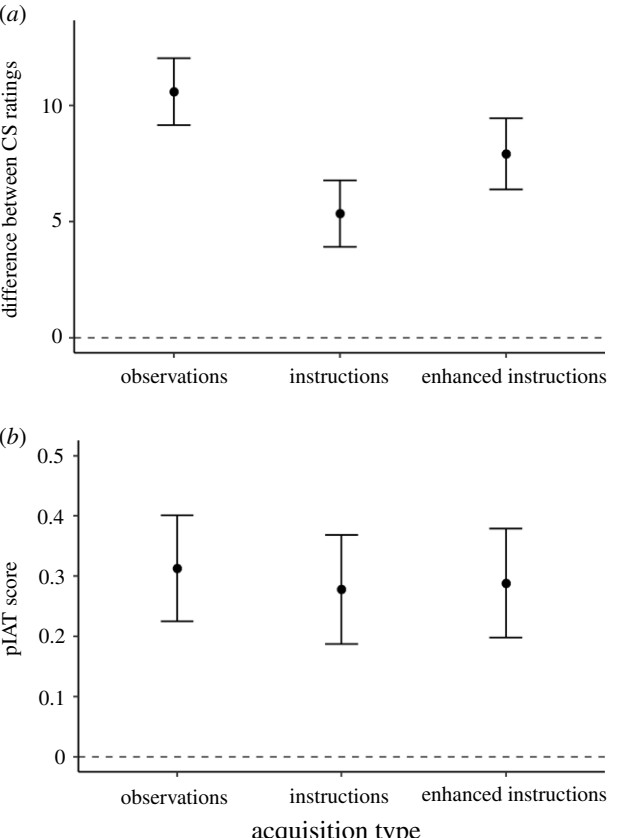

**Figure 3.** (*a*) Difference between CS$_{pos}$ and CS$_{neg}$ ratings as a function of acquisition type (Experiment 2). (*b*) pIAT scores as a function of acquisition type (Experiment 2).

## 3.3. Discussion

We replicated our prior findings insofar as both observations and instructions gave rise to self-reported changes in liking, and that observations were relatively more effective in doing so than instructions. Although enhancing the instructions (by clearly depicting what a positive or negative reaction meant) did slightly boost the effects relative to regular instructions, actual observations were still more effective. Somewhat surprisingly, this pattern was not evident on the pIAT. While all three conditions gave rise to automatic evaluations of the cookies in the expected direction, the size of pIAT scores did not differ across conditions, even though the observations and instructions conditions were identical to those used in Experiment 1. In the light of this failure to replicate the pIAT score pattern of Experiment 1, we will avoid drawing conclusions about differences between conditions from the pIAT scores in Experiments 1 and 2.

Thus for self-reported evaluations, we can conclude from Experiments 1 and 2 that (i) instructions about observations are effective in establishing likes and dislikes, (ii) actual observations seem to be more effective in doing so, and (iii) augmenting instructions through the addition of relevant information seems to slightly increase the magnitude of the effects, but still not to the same level as actually observing events for oneself. In Experiments 3 and 4, we examined if a similar pattern emerges when we look at fear rather than evaluations.

# 4. Experiment 3

We exposed participants in Experiment 3 to either an observations or an enhanced-instructions condition. Those in the observations condition encountered an observational fear conditioning procedure [42]. During the acquisition phase, they watched a sequence of videos in which a model was exposed to an unpleasant sound (delivered via his headphones) following four out of six presentations of one coloured square (CS+) but never following the presentation of another coloured square (CS−). The sound elicited clearly visible distress in the model (social US). Moreover, after his first exposure to the

sound, the model also showed anxious reactions whenever the CS+ was presented (indicating that he expected the sound to follow). The enhanced-instructions condition again showed participants examples of these reactions, followed by a description of the regularities between the CSs and the model's reactions. Note that unlike in Experiments 1 and 2, the contingency in Experiment 3 was probabilistic (i.e. the CS+ was followed by the presentation of the sound to the model on only four out of six trials, in line with recommendations for observational fear conditioning [42]). This constitutes an important difference between Experiments 1 and 2 and Experiments 3 and 4, which we will return to in the General discussion.

After the acquisition phase, all participants proceeded to a test phase. They were informed that the two CSs would now be presented on their own screen and that they might encounter an unpleasant sound through their own headphones. SCRs and self-reports of fear and sound expectancy were assessed during this phase in order to test if participants showed a stronger fear response to the CS+ than to the CS−. Importantly, no sounds were actually presented, meaning that any differences in responses to the CSs would be due to the model reactions that they were (said to be) paired with. Based on a propositional perspective, we predicted that both conditions would lead to similar changes in fear responding, with fear responses to the CS+ predicted to be larger than those to the CS−.

## 4.1. Method

### 4.1.1. Participants and design

Because it was more labour intensive to test participants in the laboratory and it was difficult to determine an effect size of interest, we opted to use a sequential Bayes factor design [43] rather than carry out a conventional power analysis. In our pre-registration, we specified that we would initially collect data for 60 participants and then calculate BFs for the differences between the two conditions. If the BFs were smaller than 1/6 (clearly favouring the null hypothesis) or larger than 6 (clearly favouring the alternative hypothesis), we would terminate data collection; otherwise, we would collect additional data from 20 participants at a time until the BFs did fall below or above these values or until we reached our specified maximum sample size ($n = 120$).

After the first 60 participants had completed the experiment, the BF for the expectancy ratings was larger than 6 and the BF for the SCRs was smaller than 1/6, while the BF for the fear ratings was still inconclusive. Unfortunately, however, the participant pool at our faculty was unexpectedly small and by the time we were able to recruit these 60 participants, we had already reached the end of the semester. Therefore, we decided to deviate from our pre-registration, terminate our data collection for Experiment 3, and instead try to replicate the initial self-report findings of Experiment 3 with a larger Prolific sample (i.e. Experiment 4).

In sum, we collected complete data for 60 participants (11 men, 49 women; $M_{age} = 22.30$, s.d.$_{age} = 4.87$). We employed a between-subjects design with two levels for acquisition type: observations versus enhanced-instructions. Stimulus assignment (whether the blue square or the yellow square served as the CS+) was counterbalanced across participants.

### 4.1.2. Materials

#### 4.1.2.1. Conditioned stimulus

A blue and a yellow square served as CSs. Both for the model in the videos as for participants during the test phase, they were presented in the middle of the screen on a black background. The videos were edited to ensure that the colours of the CSs shown in the videos exactly matched the colours of the CSs presented during the test phase.

#### 4.1.2.2. Videos

To create the impression that the model was exposed to a fear conditioning procedure over the course of several trials, we used four 17 s source videos, which were taken from a larger database shared with us by the Emotion Lab at Karolinska Institute (https://www.emotionlab.se/). The videos showed a male model who was seated in front of a computer screen and wore headphones. Each video started with a fixation cross presented on the model's screen for 2 s, after which a coloured square (CS) was presented for 6200 ms, followed by a 9 s black screen.

The behaviour of the model differed between the four source videos. In the 'unexpected sound' video, the model calmly watched the screen during the CS presentation, but showed clear discomfort as soon as the CS disappeared from screen (indicating that the unpleasant sound was presented via his headphones at CS offset). In the 'expected sound' video, the model already showed an anxious reaction as soon as the CS appeared on screen (indicating that he expected the unpleasant sound to be presented soon) and then again displayed clear discomfort when the CS disappeared from screen (indicating that the unpleasant sound was indeed presented at CS offset). In the 'omitted sound' video, the model also showed an anxious reaction when the CS appeared (indicating that he expected the unpleasant sound to be presented soon) but then visibly relaxed when the CS disappeared (indicating that no sound was actually presented at CS offset). Finally, in the 'safe' video, the model calmly watched the screen throughout the video without showing any anxiety or discomfort upon (dis)appearance of the coloured CS (indicating that he did not expect nor hear the sound).

The unexpected, expected and omitted sound videos were selected from a larger set of videos from three different models, based on pre-ratings by a separate sample of participants who were asked how they interpreted the modelled reactions at CS onset and offset (i.e. as negative and/or relieved reactions) and how believable they considered the video to be (pre-rating materials and data are available at https://osf.io/hpj9g/). All four source videos were edited to vary the colour of the CS (blue versus yellow) in order to counterbalance stimulus assignment, leading to eight videos in total. In addition, cropped versions of the unexpected and expected sound videos that only showed the model's face were created for the enhanced-instructions condition. Consequently, these two cropped videos showed only the negative reactions, not the CS that was presented on the model's screen.

### 4.1.2.3. Skin conductance measurement

SCRs were recorded using the Biosemi ActiveTwo system. Before the start of the experiment, the participant was asked to wash their hands in preparation of the placement of the skin conductance electrodes. After gently scrubbing the participant's forehead, two ground electrodes were placed approximately 3 cm apart just below the hairline. Two Ag/AgCl electrodes were attached to the thenar and hypothenar eminences of the participant's left hand. If the measured signal did not show a noticeable SCR when the participant was asked to breathe in deeply, the skin on their hand was cleaned and the electrodes were attached a second time. If the signal again showed no noticeable SCR, the experiment still proceeded but a note was made of the issue.

### 4.1.3. Procedure

#### 4.1.3.1. Preparation

After the participant received information about the experiment (which mentioned the possibility that an unpleasant sound would be presented) and provided their informed consent, the experimenter attached the SCR electrodes and placed the headphones over the participant's ears. Note that a calibration of the unpleasant sound was not included because the sound would never actually be presented during the study (similar to standard observational fear conditioning procedures, wherein electric shocks are not calibrated for the observer [42]). Finally, the experimenter explained that all necessary instructions would be provided on the screen, that the participant should move as little as possible, and that they should not speak unless they wanted to terminate the experiment (note that all instructions were provided in Dutch throughout the experiment).

#### 4.1.3.2. Acquisition phase

Figure 4 depicts the acquisition phase in both conditions. In the *observations* condition, participants were informed that they would watch videos of another participant (i.e. the model) wearing headphones and that a highly unpleasant sound could be presented through the model's headphones. Participants then watched a series of 12 videos in which the model sometimes heard the unpleasant sound after the presentation of one coloured square (the CS+) but never after the presentation of another coloured square (the CS−).

To create the impression that the CS+ was followed by the sound on four out of six trials and that the model began reacting anxiously to the CS+ after its first occurrence, the series of videos consisted of a

(*a*)

six CS+ videos (CS followed by sound in four out of six videos):

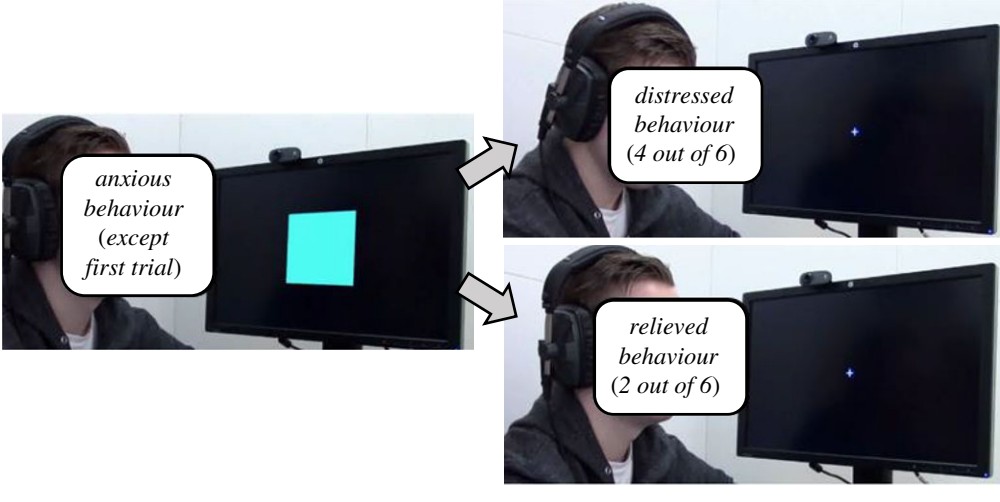

six CS-videos (CS never followed by sound):

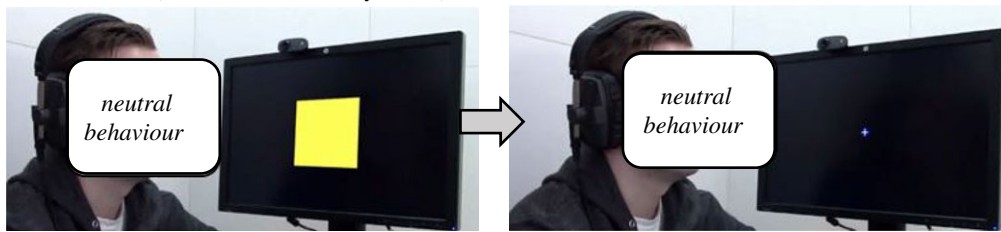

(*b*)

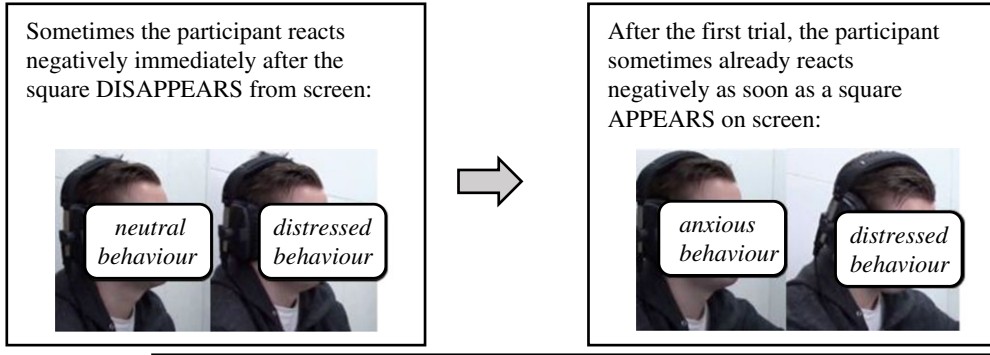

**Figure 4.** Overview of the acquisition types in Experiments 3 and 4. (*a*) Observations condition (Experiments 3 and 4) and (*b*) enhanced-instructions condition (Experiments 3 and 4). Because we do not have consent from the actor to publish images from the videos in their original form, the actor's face has been masked with labels in this figure. Naturally, this was not the case in the videos shown to participants. In the examples provided in this figure, the blue square served as the CS+ while the yellow square served as the CS−. As stimulus assignment was counterbalanced, half of the participants encountered this combination while the other half encountered the opposite combination (i.e. yellow square as the CS+ and blue square as the CS−). Please note that one condition (the information-matched observations condition of Experiment 4) is not depicted here as it only involved adding certain pieces of verbal information (see text). Figure included with permission from the Emotion Lab at Karolinska Institute (https://www.emotionlab.se/), who provided the videos depicted in the screenshots.

fixed combination of the unexpected sound, expected sound, omitted sound and safe videos (see Materials). The first two videos always showed the model reacting negatively upon the disappearance of the CS+ (i.e. unexpected sound video), while displaying no emotional reactions during a CS− trial (i.e. safe video). The remainder of the observation phase showed five more CS- trials (i.e. safe video), three trials in which the CS+ was presented and the model both expected and then actually heard the sound (i.e. expected sound video), and two trials in which the CS+ was presented and the model expected but did not actually hear the sound (i.e. omitted sound video). With the exception of the first two videos, video presentation order was random with an inter-video interval of 2 s.

In the *enhanced-instructions* condition, participants were told that later on in the experiment they would watch videos of a participant who could be exposed to an unpleasant sound via his headphones. The videos were then described to them. First, participants were told that during each video, a blue or yellow square would be presented on the model's screen and that the model might hear an unpleasant sound when the square disappeared from screen. Participants were then informed that the model would sometimes simply watch the screen, sometimes react negatively when a square disappeared from the screen (at this point, the cropped version of the unexpected sound video was played), and after the first trial sometimes also react negatively as soon as a square appeared on screen (at this point, the cropped version of the expected sound video was played).

The regularities between the CSs and the model's reactions were then described. Participants were told that during six of the 12 videos, the model would see the CS+ and would react negatively after its disappearance in four of those videos. With the exception of the very first video, the model was also said to react negatively as soon as the CS+ appeared on screen. Participants were also told that during the other six videos, the model would see the CS− and never react in a negative way. A small version of each CS was shown next to the corresponding paragraph of instructions. Participants were told to read these instructions and remember what they had been told.

### 4.1.3.3. Test phase

Participants were first told that the two CSs would now be presented on their own screen and that a sound could also be presented through their own headphones. The test phase (consisting of three blocks of trials) then began. Each trial started with the presentation of a fixation cross on a black background. After 4 s, a CS was presented, which remained on screen for 8 s, followed by an ITI of 10, 12 or 16 s (i.e. a black screen during which the sound could presumably be presented). Every block consisted of three CS+ trials and three CS− trials in a random order. After each block, participants were asked to think back to the last time they saw each CS and to indicate on scales from 1 to 9 how *anxious* they had felt (from *not at all* to *very anxious*) and to what extent they had thought the sound would be presented (from *not at all* to *very certain*). Skin conductance was measured continuously throughout the test phase. Note that no sound was actually presented.

### 4.1.3.4. Exploratory questions

After the test phase, the electrodes were detached from participants' left hand and they were asked to indicate what they believed our hypothesis was, followed by a debriefing about the absence of sounds during the test phase (and the absence of the full videos for the enhanced-instructions group). They were then asked which reaction the model had shown after each CS (or had been said to show), whether they had paid attention to the videos (or read the instructions), and whether they had paid attention throughout the test phase. Finally, they were asked how believable they considered the (cropped) videos to be, to what extent they had believed that they would see the full videos (enhanced-instructions group), to what extent they had believed a sound would be presented during the test phase, and whether their ratings had been driven by demand compliance or reactance. Participants were then thanked and debriefed.

## 4.2. Results

### 4.2.1. Data preparation

All self-report data were complete and were therefore included in the analyses. We excluded the skin conductance data of participants whose responses were not in range during (part of) the test phase ($n = 3$) or whose considerable movement during the test phase interfered with the measured signal ($n = 4$).

SCRs were calculated in the following way: after extracting the signal from a window of 2 s before CS onset until 8 s after CS onset, this signal was baseline-corrected by subtracting the mean value of the

window before CS onset. One SCR per trial was extracted by taking the maximal value of the window between 1 and 7 s after CS onset. These SCRs were then rescaled to microsiemens (μS), filtered by setting all values below 0.02 μS to zero, range-corrected by dividing each SCR by the largest SCR measured for that specific participant (to account for overall individual differences), and normalized by calculating the square root of each SCR.

## 4.2.2. Data analysis

### 4.2.2.1. Analytic strategy

Both the fear and expectancy ratings were subjected to mixed ANOVAs to test whether the ratings indeed differed between the two CSs (main effect of CS) and whether the size of this effect differed between the two groups (CS × acquisition type). We conducted another mixed ANOVA on participants' average SCRs per block to test these main and interaction effects. We also ran a linear mixed effects model on the non-aggregated SCRs (using Satterthwaite approximation for the inference tests), which included fixed effects for CS, block, condition and their interactions, to check whether the results converged with the results of the mixed ANOVA. In our pre-registration, we stated that the random effects structure of the linear mixed model would consist of only a by-participant random intercept, but following a reviewer recommendation we deviated from this plan and included all random effects supported by the design [44], meaning a by-participant intercept as well as by-participant slopes for CS, block and CS × block. This led to a singular fit, so we simplified the random effects structure until this was no longer the case (by first removing the random correlations, then removing the random slopes for the interaction and for block and then including the remaining correlation again). As the interpretation of the fixed effects did not diverge between the full and the reduced model, we report the results for the (theoretically more justifiable) full model (see [45]). For all ANOVAs, degrees of freedom and the resulting $p$-values were subjected to Greenhouse–Geisser correction if the sphericity assumption was violated. Finally, all BFs reported in this section are 'inclusion BFs', which indicate the evidence in favour of including a specific term in the model across 'matched' models (i.e. all models that did not include any interactions with the term of interest but did include the underlying main effects if the term of interest was itself an interaction term).

### 4.2.2.2. Main analyses

*Fear ratings.* Figure 5a shows the mean fear ratings as a function of CS, block and acquisition type. There was a significant main effect of CS, $F_{1,58} = 95.22$, $p < 0.001$, $\eta_p^2 = 0.62$, [0.48, 0.70], $BF_{10} > 10\,000$, such that participants reported feeling more anxious on CS+ trials than on CS− trials. The two-way interaction between CS and acquisition type was not significant, $F_{1,58} = 1.89$, $p = 0.17$, $\eta_p^2 = 0.03$, [0.00, 0.13], $BF_{10} = 1.09$, suggesting that the size of this effect did not differ between the two groups. However, there was a significant CS × block × acquisition type interaction, $F_{1.7,97.6} = 4.02$, $p = 0.027$, $\eta_p^2 = 0.06$, [0.01, 0.14] (although the BF did not support this, $BF_{10} = 0.38$). Specifically, the CS × acquisition type interaction was not significant in the first and second blocks (respectively, $p = 0.88$ and $p = 0.22$), whereas it was significant in the third block ($p = 0.005$), where the difference between CSs was smaller in the observations condition ($M = 0.53$) than in the enhanced-instructions condition ($M = 1.47$), although it remained highly significant in both groups (both $ps < 0.001$). Finally, the effects of block ($p < 0.001$), CS × block ($p < 0.001$) and block × acquisition type ($p = 0.02$) were also significant, while the main effect of acquisition type was not ($p = 0.22$).

*Expectancy ratings.* Figure 5b shows the mean expectancy ratings as a function of CS, block and acquisition type. There was a significant main effect of CS, $F_{1,58} = 109.87$, $p < 0.001$, $\eta_p^2 = 0.65$, [0.53, 0.73], $BF_{10} > 10\,000$, such that participants expected a sound more after the CS+ than after the CS−. Importantly, there was a significant interaction between CS and acquisition type, $F_{1,58} = 7.54$, $p = 0.008$, $\eta_p^2 = 0.12$, [0.02, 0.25], $BF_{10} = 264.33$, such that the overall difference between CSs was smaller in the observations condition ($M = 1.53$) than in the enhanced-instructions condition ($M = 2.62$). However, it was highly significant in both groups (both $ps < 0.001$). Other than effects of block ($p < 0.001$) and CS × block ($p < 0.001$), there were no other significant effects (acquisition type: $p = 0.13$; block × acquisition type: $p = 0.24$; CS × block × acquisition type: $p = 0.31$).

*SCRs.* Figure 5c shows the mean SCRs as a function of CS, block and acquisition type. There was a significant main effect of CS, $F_{1,51} = 20.08$, $p < 0.001$, $\eta_p^2 = 0.28$, [0.12, 0.43], $BF_{10} > 10\,000$, such that SCRs were larger during CS+ presentations than during CS− presentations. It did not vary as a function of acquisition type, $F_{1,51} = 0.43$, $p = 0.52$, $\eta_p^2 = 0.01$, [0.00, 0.09], with the BF ($BF_{10} = 0.13$) suggesting similar fear learning in the two conditions. Other than a main effect of block ($p < 0.001$), there were

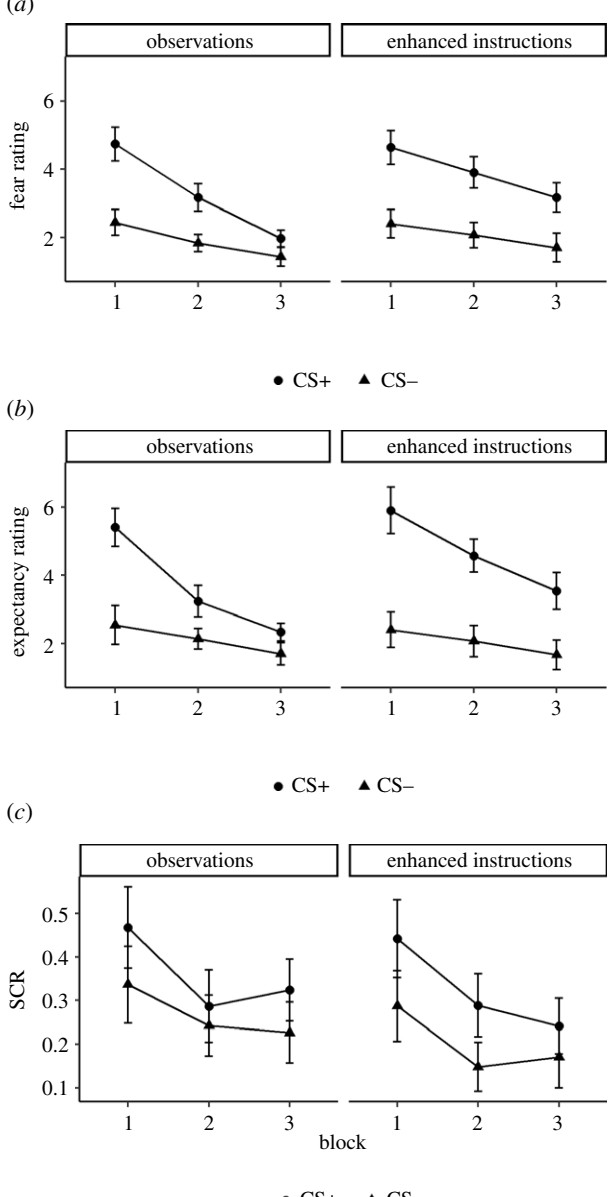

**Figure 5.** (*a*) Fear ratings as a function of CS, block and acquisition type (Experiment 3). (*b*) Expectancy ratings as a function of CS, block and acquisition type (Experiment 3). (*c*) SCRs as a function of CS, block and acquisition type (Experiment 3).

no other significant effects (CS × block: $p = 0.39$; acquisition type: $p = 0.22$; block × acquisition type: $p = 0.85$; CS × block × acquisition type: $p = 0.41$). Finally, the results of the linear mixed effects analysis on the non-aggregated SCRs were highly similar, showing only significant effects of CS, $F_{1,56.8} = 19.71$, $p < 0.001$, and block, $F_{2,67.6} = 10.59$, $p < 0.001$.

### 4.2.2.3. Exploratory analyses

A few points are worth mentioning (see the electronic supplementary material for all exploratory analyses). First, far fewer participants in the observations condition correctly recalled that the CS+ was sometimes followed by the model's negative reactions (17 participants as opposed to 29 participants in the enhanced-instructions condition). When only participants who had perfect memory for the regularities were included ($n = 44$), the main results were largely unchanged, except that by the third block, the observations group no longer showed an effect in terms of fear ratings. Second, our main results were unchanged when we only included participants who reported no demand compliance ($n = 57$). When we only included participants who reported no reactance ($n = 43$), the CS × block × acquisition type effect on the fear ratings was no longer significant.

## 4.3. Discussion

Experiment 3 moved from evaluations to fear and examined if observations would once again give rise to relatively larger effects than instructions about observations. Results indicated that both observations and instructions about observations led to a change in behaviour on all measures (fear ratings, expectancy ratings and SCRs). Unlike what we found for evaluations, the effects of actual observations were not larger than those generated via enhanced instructions about observations. If anything, evidence suggested that the latter had a larger impact on expectancy ratings and (to some extent) fear ratings.

Given the results of Experiments 1 and 2, it may seem somewhat surprising that the enhanced instructions had slightly larger effects than the observations in Experiment 3. One possible explanation may be that participants in the enhanced-instructions condition more accurately remembered which CS was followed by negative reactions than those in the observation condition (i.e. they had better contingency memory). The current pattern of findings could also have been due to the fact that the enhanced-instructions group was explicitly told that a sound could be presented to the model at CS *offset*, that *only one* of the CSs would be followed by negative reactions, and that they needed to *remember* which. The observation group received no such instructions.

# 5. Experiment 4

Because of these differences between the two conditions in Experiment 3, and the smaller than expected sample size in that experiment, we decided to conduct a follow-up experiment with a larger sample to see if the stronger impact of enhanced instructions (relative to observations) on expectancy and fear ratings would replicate. We additionally included an 'information-matched observations' condition, that is, an observations condition that did include the above pieces of information and thus was more directly comparable to the enhanced-instructions condition. If these pieces of information were indeed responsible for the pattern observed in Experiment 3, we would expect (i) a larger effect in the enhanced-instructions condition than in the observations condition, replicating Experiment 3, (ii) a larger effect in the information-matched observations condition than in the standard observations condition, and (iii) similar effects in the enhanced-instructions and information-matched observations conditions.

Experiment 4 also contained a number of other changes. First, and most importantly, we collected the data via Prolific Academic, which allowed us to recruit a much larger sample of participants. Although this naturally reduced the experimental control that we could exert over participants' behaviour, we gave clear instructions about the need to use their headphones and the volume they should set (see below), and we asked at the end whether they had followed these instructions or not. Collecting the data online also meant that we could no longer measure physiological responses. Second, we considerably shortened the test phase and asked participants to rate their fear and expectancy after every trial rather than in a blocked fashion. Finally, because participants in Experiment 3 often seemed to interpret the videos in a different way than we had intended (e.g. assuming that the model was also exposed to a sound whenever he showed an apprehensive reaction at CS onset), we asked a number of exploratory questions about how they interpreted the videos.

## 5.1. Method

### 5.1.1. Participants and design

Based on a power analysis indicating that we required a total sample size of $n = 189$ in order to have 80% power to detect an effect size of $\eta_p^2 = 0.03$ (the smaller of the two relevant effect sizes obtained in Experiment 3) in a mixed ANOVA, we planned to recruit a sample of $n = 192$ participants. As only a small percentage of participants reported not following all of the instructions with regard to the headphones and sound settings, we collected data until we had complete data for 192 participants who reported following all of these instructions (no other data analysis or exploration was performed in the meantime). The total sample consisted of 211 participants (138 men, 71 women, 1 non-binary person, 1 person whose gender was not recorded properly; $M_{age} = 27.05$, s.d.$_{age} = 6.89$).

We used a between-subjects design with three levels for acquisition type: observations, information-matched observations and enhanced-instructions. Stimulus assignment was again counterbalanced across participants.

### 5.1.2. Materials

The CSs and videos were identical to those used in Experiment 3.

### 5.1.3. Procedure

#### 5.1.3.1. Introduction

After providing their informed consent, participants were instructed to plug in their headphones, set the volume on their computer to 20% and wear their headphones throughout the study. They were also informed that during one of the tasks a highly unpleasant sound would be presented to them from time to time.

#### 5.1.3.2. Acquisition phase

The manipulations in the *observations* and *enhanced-instructions* conditions were largely identical to those in Experiment 3, with the exception that on instructions pages participants could not proceed until a certain amount of time had passed (in order to maximize the probability that they read all instructions).

In the *information-matched observations* condition, participants watched the same series of videos as the observations group. However, in order to make this condition as comparable as possible to the enhanced-instructions condition (i.e. 'matched' in terms of the information that was communicated), they were informed that (i) when a square *disappeared* from screen, a very unpleasant sound might be presented to the model, (ii) the model might react in a negative way after *one* of the squares, and (iii) they had to *remember* which coloured square went together with the negative reactions because they would need this information later on.

#### 5.1.3.3. Test phase

The test phase consisted of eight trials (four CS+ and four CS− trials). Each trial started with a black screen presented for 1.5, 2 or 2.5 s, followed by a fixation cross presented for 2 s. Next, the CS was presented and stayed on screen for 6 s, followed by a black screen presented for 5, 6 or 7 s (during which the sound could presumably be presented). After each trial, participants were asked to rate on scales from 1 to 9 how anxious they felt when they saw the coloured square (from *not at all* to *very anxious*) and to what extent they had thought the unpleasant sound would be presented (from *not at all* to *very certain*). Once again, no sounds were actually presented.

#### 5.1.3.4. Exploratory questions

Participants received an initial debriefing about the absence of sounds during the test phase (as well as the absence of full videos for the enhanced-instructions group). To assess whether they had interpreted the videos (or our description thereof) as intended, they were then asked when (i.e. during, after or both during and after the presentation of a CS) the model could hear the unpleasant sound, after which CS the model had shown (or been said to show) negative reactions, and how often that CS had been followed by the sound to the model (i.e. never, sometimes or always). Importantly, given the online setting, they were also asked whether they had followed all of our instructions (i.e. plugged in and wore their headphones, as well as set their sound to 20%), and if not, what they had not done and why. In order to encourage participants to respond honestly to these questions, it was explicitly stated that their answers would not affect the payment they would receive for their participation and that it was very important that they answered truthfully. They also answered the same questions as in Experiment 3, after which they were thanked and debriefed.

## 5.2. Results

### 5.2.1. Data preparation

We first excluded the data of participants who provided incomplete data or who reported technical issues ($n = 18$), as well as one participant who was not fluent in English. This resulted in complete data for 211 participants. We also excluded the data of one participant who had read the instructions twice. Finally, we excluded 20 participants who reported not following all of our instructions with regard to their headphones and sound settings (six reported not using headphones, eight reported not

setting their volume to the specified value, four reported both not using headphones and not setting the specified value, and two reported taking off their headphones during the test phase). Our final sample consisted of 190 participants (129 men, 59 women, 1 non-binary person, 1 person whose gender was not recorded properly; $M_{age} = 27.05$, s.d.$_{age} = 6.82$).

Differential ratings were calculated by subtracting the (fear or expectancy) rating on the first CS– trial from the corresponding rating on the first CS+ trial, subtracting the rating on the second CS– trial from the corresponding rating on the second CS+ trial, and so on.

### 5.2.2. Data analysis

#### 5.2.2.1. Analytic strategy

Both the differential fear and expectancy ratings were subjected to mixed ANOVAs to test whether they were significantly larger than zero overall (i.e. the intercept), whether they changed across the test phase (i.e. a main effect of trial), and whether they varied as a function of acquisition type (i.e. a main effect of acquisition type). We then conducted pairwise comparisons to test which of the three groups differed from each other, using Holm–Bonferroni correction to account for multiple comparisons. Corrections for sphericity violations were applied as in Experiment 3 and BFs again reflect the evidence for including a specific term in the model.

#### 5.2.2.2. Main analyses

*Fear ratings.* Figure 6a depicts the fear ratings as a function of CS, trial and acquisition type. The intercept was significantly larger than zero, indicating that participants reported more fear after the CS+ than after the CS–, $F_{1,187} = 206.47$, $p < 0.001$, $\eta_p^2 = 0.52$, [0.44, 0.59]. Crucially, the differential fear ratings varied as a function of acquisition type, $F_{2,187} = 4.84$, $p = 0.009$, $\eta_p^2 = 0.05$, [0.01, 0.10], $BF_{10} = 4.53$. Specifically, the effect in the information-matched observations condition ($M = 2.61$) was larger than the effects in the observations condition ($M = 1.77$), $p = 0.03$, $BF_{10} = 2.78$, and the enhanced-instructions condition ($M = 1.63$), $p = 0.01$, $BF_{10} = 6.77$. The latter two conditions did not differ, $p = 0.68$, $BF_{10} = 0.21$. Finally, the effect of trial was significant, $p = 0.018$, while the interaction between acquisition type and trial was not, $p = 0.19$.

*Expectancy ratings.* Expectancy ratings were also higher for the CS+ relative to the CS– (figure 6b), as reflected by the intercept, $F_{1,187} = 217.46$, $p < 0.001$, $\eta_p^2 = 0.54$, [0.46, 0.60]. Similar to the fear ratings, the size of this effect varied as a function of acquisition type, $F_{2,187} = 6.18$, $p = 0.003$, $\eta_p^2 = 0.06$, [0.01, 0.12], $BF_{10} = 12.7$. Once again, the effect in the information-matched observations group ($M = 3.02$) was larger than the effect in the observations group ($M = 2.15$), $p = 0.047$, $BF_{10} = 1.78$, as well as larger than the effect in the enhanced-instructions group ($M = 1.71$), $p = 0.002$, $BF_{10} = 29.84$, while the observations and enhanced-instructions groups did not differ, $p = 0.25$, $BF_{10} = 0.37$. The main effect of trial was significant, $p < 0.001$; the acquisition type × trial interaction was not, $p = 0.20$.

#### 5.2.2.3. Exploratory analyses

A number of points are worth mentioning (see the electronic supplementary material for all exploratory analyses). First, the information-matched observations group showed the best memory in terms of *which* CS was (not) followed by negative reactions. In line with the possibility that memory for the pairings might have been partially responsible for differences between groups, the main effect of acquisition type on the fear ratings became non-significant when participants with imperfect contingency memory ($n = 32$) were excluded. Second, participants' answers suggested that a substantial number of them did not interpret the videos as we had intended. Specifically, while the majority in the enhanced-instructions group (correctly) indicated that the CS+ was only *sometimes* followed by the sound to the model, the majority of the other two groups indicated that the CS+ was *always* followed by the sound to the model. Similarly, the majority of the enhanced-instructions group also correctly reported that the sound was presented to the model at CS offset, while relatively more participants in the other two groups believed that the model could hear a sound at CS onset, or at both CS onset and offset. Finally, most participants believed that a sound would be presented during the test phase, suggesting that our instructions with regard to the sound were still believable in the online setting.

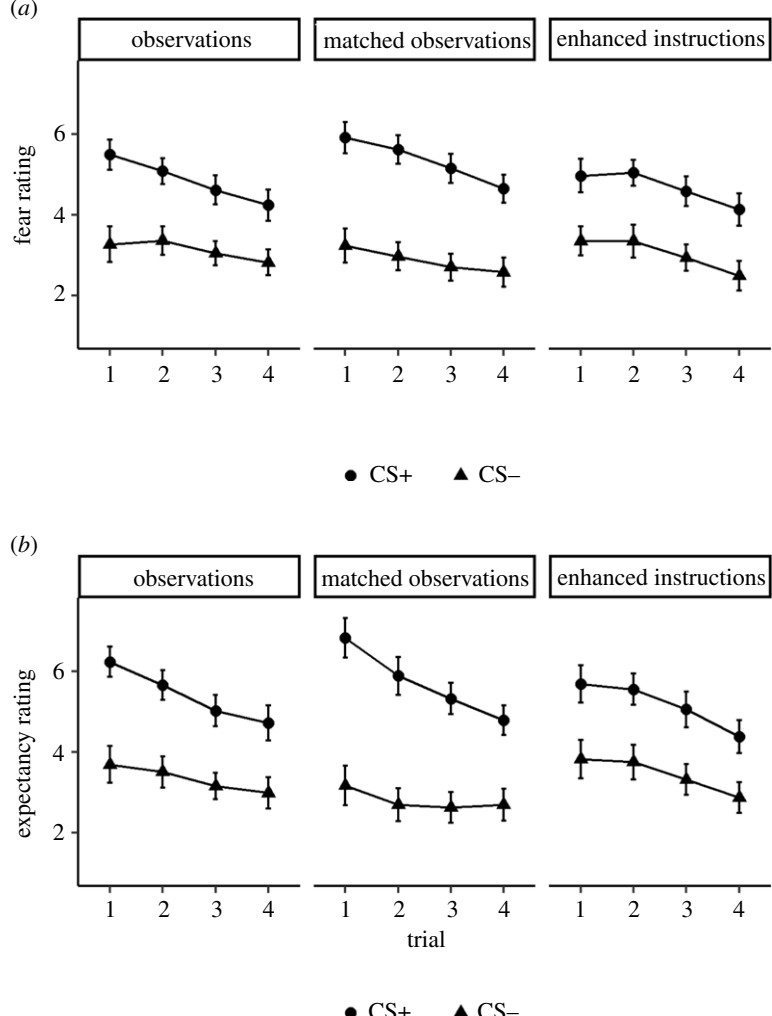

**Figure 6.** (a) Fear ratings as a function of CS, trial and acquisition type (Experiment 4). (b) Expectancy ratings as a function of CS, trial and acquisition type (Experiment 4).

## 5.3. Discussion

Once again, both observations and instructions about those observations had a clear impact on fear and expectancy ratings. Despite using identical enhanced-instructions and observations manipulations as in Experiment 3, we did not replicate the finding that enhanced instructions led to larger effects. Instead, the two groups produced effects of comparable size. However, when the observations condition was augmented with the additional information included in the enhanced-instructions condition (i.e. the information-matched observations condition), a slight increase in effects emerged. That said, the evidence for the differences between the two observations conditions was anecdotal at best (based on the Bayes factors) and these differences became non-significant when participants who had incorrect contingency memory or reported reactance were excluded (see electronic supplementary material).

The comparison between the enhanced-instructions condition and the information-matched observations condition is perhaps the most relevant of all, as these two groups received the same pieces of information (i.e. the content presented in the two conditions was equated) but in a different format. The results of this comparison suggested that observations had a stronger effect than instructions about those observations.

Taken together, we can conclude based on the results of Experiments 3 and 4 that instructions about observations (i.e. about the relation between stimuli and a model's behaviour) consistently resulted in fear learning, both in terms of self-reports as well as for a physiological measure of fear (SCRs). Whereas Experiment 3 suggested that instructions may have a larger effect on self-reports than observations, this pattern was not replicated in Experiment 4, and a more direct comparison suggested that observations may actually have a larger impact on behaviour.

# 6. General discussion

Past conditioning research provided extensive evidence for several learning pathways: the experience of CS–US pairings, instructions about CS–US pairings, and the observation of a model's behaviour in the presence of a stimulus. Whereas previous research on instructions generally focused on the impact of information about CS–US pairings that would be experienced directly, we examined for the first time whether instructions about the behaviour of a model in the presence of a stimulus would lead to changes in behaviour, more specifically evaluative (Experiments 1 and 2) and fear responding (Experiments 3 and 4). We also compared the relative power of observations versus instructions about observations in triggering behavioural changes. In doing so, we explored whether differences in the impact of instructions and observations relate to differences in the information that those events convey. Finally, in Experiment 1, we also explored whether there would be an additive effect of actual observations on top of receiving instructions about observations.

The question of whether instructions about observations influence behaviour can be answered unequivocally: we obtained clear effects of instructions about observations in all of our experiments and for all measures of evaluative and fear responding (self-reported, automatic and physiological). In other words, simply telling someone about a regularity between a stimulus and the behaviour of a model can suffice to change their behaviour.

When it came to the relative effectiveness of instructions about observations versus actual observations, no clear picture emerged. In some cases, instructions were as effective as observations. Yet most of the time observations led to slightly larger effects than instructions. In addition, observing the regularities after already receiving instructions still had an additive impact on evaluations in Experiment 1.

Throughout our research, we also endeavoured to increase the 'match' between the two pathways in terms of the information they conveyed. In Experiment 2, we added examples of the model's reactions to the instructions, thereby matching more closely the information conveyed by actual observations. Interestingly, adding information about the nature of the model's reactions slightly increased the impact of those instructions on evaluative ratings. In Experiment 4, we added information to the observations that was given only to the enhanced-instructions group in Experiment 3, namely that there would be a regularity and that participants had to remember it. This also slightly increased the effects of the observations on fear responding, which highlights the potential impact of mentioning the presence and importance of a regularity (for a related finding, see [46]). Although these findings should be interpreted with caution (the Bayes factors indicated only anecdotal evidence), they suggest that subtle differences in the provided information may influence the relative effectiveness of learning pathways and underline the importance of closely matching this information if one wants to compare different pathways.

## 6.1. Theoretical considerations

As noted in the introduction, cognitive models of learning differ in the way that they account for the effects of observations and instructions. In this section, we discuss how our results may inform the available theoretical perspectives. From a propositional perspective (e.g. [31]), one would predict the effects to be similar if the information that is conveyed by actual events and instructions about events is similar. In line with this idea, differences between groups in our studies seemed to be partially due to 'mismatches' in terms of the information they were given, such as information about the precise nature of the model's reactions or about the presence of an important regularity between the reactions and the CSs. Both of these are aspects that would be predicted to influence the effectiveness of the learning pathways from a propositional perspective.

On the other hand, even when the information was closely matched (Experiments 2 and 4), observations still seemed to have stronger effects than instructions about observations. How might one explain these findings? A first possibility is that learning via instructions and learning via repeatedly experienced or observed regularities depend on partially distinct (neuro)cognitive processes and therefore do not necessarily have similar effects on behaviour, as assumed by dual-process perspectives (e.g. [8,34,35]). Note that while some specific dual-process theories cannot easily accommodate the effects of instructions on automatic evaluations found in Experiments 1 and 2 without additional assumptions (e.g. [34]), other theories can (e.g. [36]). Generally speaking, however, the mixed findings that we obtained seem to fit well with the idea that multiple processes are involved.

A second possibility is that the two pathways rely on the same processes but differ in other respects, which could account for at least some of the discrepancies observed in our data. What might those differences be? First, and most trivially, some of the differences observed in our studies may have been driven by differences in attention. Especially in online studies, participants may read instructions quickly and superficially, whereas the use of videos may increase attention and engagement with the experiment. This may also explain why we observed larger effects of instructions only in Experiment 3, where participants completed the experiment in the laboratory and were more likely to take the time to read every page of instructions.

Second, the instructions and observations might differ in terms of believability. For example, the impact of instructions about observations depends on participants believing that the described events actually occurred, whereas they can see the events 'with their own eyes' when they watch the videos. Unlike the previous point about attention, believability may constitute a more intrinsic difference between pathways as it seems difficult to fully match the believability of the information. Nevertheless, future research could test whether differences between pathways are related to differences in reported believability of those pathways. Likewise, differences between pathways might be reduced by reducing differences in believability (e.g. by adding a pre-training phase in which instructions are proven to be accurate). Also note that believability could in principle influence the relative effectiveness of pathways in either direction. For example, if the model's behaviour seems acted or exaggerated, believability may actually be lower when one watches a video relative to when one is simply told that the model reacted in a certain way, leading to a smaller effect of observations relative to instructions.

Third, in our studies (as well as in many other learning studies), learning might depend on the successful transmission of several components of information. A first component is the (strength of the) regularity between events (i.e. which stimulus is followed by which reaction and how often). A second component is the nature of the events themselves (i.e. the precise nature and intensity of the model's reaction, which may in turn be used to infer the properties of the CS). There may be differences in terms of how well pathways can convey either component.

On the one hand, instructions may be more suited for communicating the first component, because they can clearly and directly describe the regularities as well as their strength, avoiding the need for participants to 'figure out' the regularities themselves. In line with this idea, instructions have been found to be more effective in changing the evaluative responses of children [26]. When discussing this component, we should also highlight that our experiments contained two different types of regularities. In Experiments 1 and 2, the regularities were deterministic (i.e. the $CS_{pos}$ was followed by a positive reaction and the $CS_{neg}$ was followed by a negative reaction). Therefore, after receiving the instructions or after seeing one video of each reaction, participants no longer needed to update their representation of the regularities (i.e. figure them out). In Experiments 3 and 4, however, the regularity was probabilistic (i.e. the CS+ was followed by a distressed reaction four out of six times). While participants who received instructions were informed directly about the strength of this regularity, participants in the observations conditions needed to discover this strength over the course of the entire observation phase (in computational terms, the learned value of the CS+ continuously needed to be updated based on error prediction across several trials; for recent discussions of (probabilistic) social learning from a computational perspective, see [37,47]). As a result, the observations conditions in Experiments 3 and 4 involved considerably more 'figuring out' than the observations conditions in Experiments 1 and 2 or the instructions conditions in any of our experiments.[5] This probably left more room for error. Some of the findings from Experiment 4 seem to support the notion that accurately encoding the strength of the regularity was more challenging in the observations conditions. Specifically, while participants in the information-matched observations condition accurately remembered which CS was followed by which reaction, they actually *overestimated* the strength of the regularity, reporting that the CS+ was always followed by the unpleasant sound (whereas in reality the model showed distress only after some presentations of the CS+). In other words, it seems that the videos were less suited to convey the imperfect contingency than the instructions, which may have led to stronger, but less accurate, fear learning. Finally, although the contingency was deterministic in Experiments 1 and 2, some participants in the observations condition may have mistakenly interpreted the repeated videos as showing three different instances of the model tasting each cookie, rather than showing the same instance three times. If so, this would constitute an important difference with the instructions condition from a

---

[5]We thank an anonymous reviewer for drawing our attention to this important point.

propositional perspective (whereas only the number of repetitions and the strength of the resulting association would seem to matter from a purely associative perspective). Specifically, if some participants believed that the model consistently reacted a certain way to a CS, this may have increased the extent to which they attributed the model's reaction to an inherent property of the CS (e.g. see [48]), which may in turn have increased the size of effects. Therefore, it would be informative for future research to assess in a more fine-tuned way what exactly participants believe they have observed (e.g. single or multiple reactions), as their perception of the strength of the relation may vary accordingly.

On the other hand, observations may be more suited for conveying the second component (i.e. the nature of the events themselves): a few seconds of video can show a richness of dynamic facial expressions and body language that may be difficult to put into words. In line with this idea, providing participants with an example of the model's reaction prior to the instructions seemed to increase the similarity of the effects to those found in the observations group.

The idea that certain types of information are better conveyed by certain pathways is interesting, because it implies that which pathway has the most powerful impact on behaviour depends on the specific information that is conveyed. Moreover, as we hinted at above, the fact that a pathway is more suited for conveying a certain piece of information does not mean that the resulting change in behaviour would necessarily be stronger. If a video successfully conveys to an observer that a model's reaction is very subtle and not at all intense in nature, observations may actually lead to smaller effects than instructions describing that reaction with a few words. Conversely, instructions may have smaller effects than observations if they successfully convey that a regularity is not very strong.

In sum, it is possible that learning via observations and learning via instructions are mediated by different cognitive processes, and some of our findings seem in line with this idea. However, because of inherent differences in the way these two pathways can convey information, their effects may differ even if the underlying process is the same. From this latter perspective, it may not be very interesting to try and determine which pathway is 'more powerful' in some absolute sense, as this may heavily depend on the type of information that is communicated and on the context. Instead, it may be more useful to explore the conditions under which the different pathways have a larger impact on behaviour.

## 6.2. Limitations and future directions

Our research has a number of limitations which can inform future research on this topic. First, most of our conclusions about differences between groups are based on self-reports. Although we did include a measure of automatic evaluations (pIAT), the conflicting findings from Experiments 1 and 2 prevent us from drawing any conclusions regarding the comparison between the two pathways in terms of automatic evaluative responding. We also included a physiological measure of fear in Experiment 3 (i.e. skin conductance). Although it was encouraging to see an effect of instructions about observations on this measure as well, the sample size in this study was rather small and thus statistical power to detect differences between the two groups was relatively low. Future research on the comparison between observations and instructions could therefore include (other) measures of automatic and physiological responding, especially if one aims to test cognitive theories that predict discrepancies between these measures and self-reports, as is the case for some of the theories discussed in the introduction.

Second, we have focused on differences between conditions in terms of the learning effects averaged across participants. However, distribution plots (see the electronic supplementary material) suggest that the means may not always reflect the full picture. For example, in both evaluative learning experiments, there were some participants who showed no learning effect at all on self-reports, especially in the instructions condition. For practical purposes, it seems relevant to keep this in mind: if a learning procedure has a strong impact on a few participants but no impact at all on the majority, only looking at the mean could suggest that the procedure changes behaviour, while it may have limited practical value (see also [49]).

Third, based on participants' responses to the exploratory questions, the videos in our studies were acceptable but rather limited in terms of believability. As we have discussed above, believability is likely to play a role in how strongly participants are influenced by the learning procedure. Therefore, future research could attempt to increase the believability of videos used in the observational learning task, or even use live modelling instead of videos (although this may also increase ambiguity and thus reduce learning [50]).

Fourth, the instructions provided to participants in Experiments 1 and 2 were very minimal, merely describing the model's behaviour as 'positive' or 'negative'. This left much room for participants' own assumptions in terms of how strong the reactions would be, which we tried to reduce by showing an example in Experiments 2 and 4. Because such assumptions and the resulting mental imagery regarding the precise nature of the model's behaviour are likely to have an impact, future research could (i) look at the role of imagery (e.g. by measuring participants' general tendency to imagine events vividly or actively asking participants to imagine the reactions) and (ii) investigate if it is possible to achieve equally large effects of a purely verbal manipulation if the model's reactions are described in a very detailed manner, similar to how someone's behaviour may be described in a work of fiction.

Fifth, Experiments 1 and 2 involved food-related preferences, which may be highly subjective. From a propositional perspective, whether participants assume that they are likely to share the model's preferences probably plays an important role in determining their own preferences. Because this consideration applies equally to the effects of both actual observations and the effects of instructions about those observations, we consider it unlikely that the groups in our experiments differed with regard to this assumption. Nevertheless, it would be interesting to include an exploratory question about it in future studies, as this may provide more insight into the determinants of participants' responses.

Finally, we attempted to make the information contained within the different pathways as similar as possible within the boundaries inherent to the different formats. Although this certainly created challenges, we believe that future research aimed at comparing pathways could benefit from trying to match the information as closely as possible. Especially if one wishes to make claims about the cognitive processes that mediate the different pathways, it is crucial that the compared manipulations contain the same information (*content*) and differ only in terms of the *format* in which it is presented.

# 7. Conclusion

To our knowledge, our research constitutes the first attempt to examine the effects of instructions about observations and directly compare the effects of observing regularities between stimuli and the behaviour of a model (observational conditioning) to the effects of receiving a verbal description of those same events. We consistently found that both pathways (observations and instructions) were effective in changing what people fear as well as what they like and dislike. Yet observing regularities for oneself led to larger effects in some (but not all) of the experiments. Although this could reflect different cognitive processes mediating the two learning pathways, another possibility is that the pathways differ in terms of believability and in how suitable they are for conveying certain pieces of information (e.g. the nature and strength of a relation between events versus the exact nature of the events). Therefore, it may be more interesting to investigate the conditions under which each pathway has the strongest impact on behaviour rather than try to make judgements about which pathway is more effective in some absolute sense.

Ethics. All experiments were conducted after obtaining advice and approval from the Ethical Committee of the Faculty of Psychology and Educational Sciences (application no. 2018/53). All participants provided informed consent.
Data accessibility. All datasets and analysis code supporting this article are available on the Open Science Framework (http://dx.doi.org/10.17605/OSF.IO/AY25Z) [51]. With the exception of the videos, which have been replaced by anonymized versions because we did not have consent from the actors to publish them in their original form, all research materials are also available.

Additional figures and results are provided in the electronic supplementary material [52].
Authors' contributions. S.K.: conceptualization, data curation, formal analysis, funding acquisition, investigation, methodology, project administration, resources, software, visualization, writing—original draft and writing—review and editing; S.H.: conceptualization, funding acquisition, methodology, software, supervision and writing—review and editing; J.D.H.: conceptualization, funding acquisition, methodology, resources, supervision and writing—review and editing.

All authors gave final approval for publication and agreed to be held accountable for the work performed therein.
Competing interests. We have no competing interests.
Funding. S.K. is supported by PhD fellowship FWO18/ASP/119 from the Research Foundation Flanders (FWO) to S.K. S.H. and J.D.H. are supported by grant no. BOF16/MET_V/002 from Ghent University to J.D.H.
Acknowledgements. We would like to thank the members of the Emotion Lab (led by Andreas Olsson) at the Karolinska Institute for providing us access to their database of observational fear conditioning videos, and Sophie Wüstefeld for

collecting part of the data for Experiment 3. We also thank three anonymous reviewers for their helpful comments on an earlier draft of the paper.

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
