## [Peer Review File · Royal Society Open Science]

Review History

RSOS-211085.R0 (Original submission)

Review form: Reviewer 1

Is the manuscript scientifically sound in its present form?

Yes

Are the interpretations and conclusions justified by the results?

No

Is the language acceptable?

Yes

Do you have any ethical concerns with this paper?

No

Have you any concerns about statistical analyses in this paper?

No

Recommendation?

Reject

Comments to the Author(s)

In their manuscript "Learning via instructions about observations: Exploring similarities and differences with learning via actual observations" the authors report a set of experiments which documents the effect of "instructions about observations" on various affective learning indices. The basic finding is interesting - people can learn through hearing about other people's experiences - , but hardly surprising given how well documented various instructional learning effects are. Nonetheless, the experiments are carefully conducted and presented in a balanced manner. However, the theoretical framework is, in my view, weak, and in the end its not entirely clear what we have learned (except that people learn via hearing about others experiences). Rather than testing substantial hypotheses, the paper has a "its known that $1+1 = 2$, and $2+2 = 4$, but what about $1+2$?" - character. In other words, it documents a (expected) phenomenon. In my opinion, the paper would benefit from acknowledging this, rather than claiming to test theory-derived hypotheses.

My main issue is that the authors preferred "propositional perspective" is so vague that evaluating the logic of predictions/hypotheses is impossible. The main prediction seems to be "to the extent that actually observed events and instructions about those observed events result in the same or a similar proposition being formed and considered valid, this perspective predicts that both pathways would have a similar impact on behaviour." From this it's unclear both (i) what propositions are, and (ii) how the authors will measure whether similar propositions are formed. Both issues could be addressed with a formal model that made it crystal clear what associations are, and make clear a priori assumptions about the validity of propositions.

Instead, the authors seem to find this level of unspecific reasoning sufficient. As consequence, it's very unclear what theoretical perspective the results support (as acknowledged in the discussion). The "propositional perspective" seems, due to its lack of specificity, nearly impossible to falsify. In the face of data that shows differences between different types of social information (which is not predicted by the propositional perspective by definition), the authors try to salvage it by a number of axillary hypotheses (pages 42-43).

For example, they propose that the number of pairings and the strength of reactions to stimuli could explain differences between "pathways". Why would a propositional mechanism care about such things? If learning is governed solely by the truth value of propositions, a single piece of information should be as efficient as multiple pieces. Or do the authors assume that truth value increases with repetitions? If so, the authors could consider using established and precise Bayesian learning models that implement exactly such belief updating and allows quantitative predictions.

What do other theories predict? The authors state "Based on a propositional perspective, we predicted that all three conditions would have a similar impact on evaluations of the cookies (with the CSpos being evaluated more positively than the CSneg)." But such statements are of course more meaningful if different theories make different apriori predictions.

For experiment 2, the authors state ""Based on a propositional perspective, we would predict that the effects in this enhanced-instructions condition would be (a) larger than the effects in the instructions condition and (b) similar in magnitude to the effects in the observations condition (given that the information they conveyed was now closer in nature)." Why would any additive effects be expected if propositions represent the truth value of relationships in the world?

In summary, these issues highlight the need for a formal specification (c.f. reinforcement learning and Bayesian learning) of the “propositional perspective” for it to be taken seriously as an account of learning.

Minor.

- The concept of pathways is not well defined. Why would instructions about events befalling a third party represent a different pathway than instructions about events happening to oneself?
- The statistical models are sometimes rather mindlessly applied. Specifically, the authors report two ANOVA analyses with rather silly results (p 15): “The main effect of acquisition type was qualified by an interaction with stimulus assignment, $F(2, 260) = 3.51, p = .03$, such that the effect was only significant if Plogo served as the CSpos” and “The main effect of acquisition type was qualified by an interaction with block order, $F(2, 262) = 3.31, p = .038$, such that it was only significant if participants completed the learning-consistent block first.” These results seem quite obviously spurious, and one wonders why these factors were included in the model. If taken seriously, the authors should discuss the limits to generalization.
- Was the US in Experiment 3 calibrated for each subject?
- Please specify the linear mixed-effects model (p 28)
- What evidence is there that participants in experiment 4 truthfully reported whether they wore headphones etc? In other words, why would participants be truthful in their reports? Did the authors safeguard against participants thinking that they might lose monetary rewards if truthful?

Review form: Reviewer 2

Is the manuscript scientifically sound in its present form?

Yes

Are the interpretations and conclusions justified by the results?

Yes

Is the language acceptable?

Yes

Do you have any ethical concerns with this paper?

No

Have you any concerns about statistical analyses in this paper?

No

Recommendation?

Major revision is needed (please make suggestions in comments)

Comments to the Author(s)

In this paper, Kasran and colleagues investigate how getting instructions about another person's responses influences learning versus actually observing their responses. The authors report that both instructions about observation and actual observations are effective in influencing the

learner's own evaluative (Experiments 1-2) and fear (Experiments 3-4) responses. Actual observations were more effective than instruction in some cases but not all.

This study provides valuable data to the field of social learning. As the authors point out in the introduction and discussion, the strength of behavioral change can vary depending on the pathways (in this case, actual observations vs instructions about observations) and the type of information being conveyed, and there are practical implications in finding out which ones are effective in which scenario. However, the theoretical implications are rather unclear to me. I will elaborate it below with some additional questions and comments.

1. What would be the theoretical implication for showing the effectiveness of instructions about observations (e.g., "A model would react negatively to an unpleasant sound that followed a yellow square") as opposed to instructions about the pairing itself (e.g., "An unpleasant sound will be played following a yellow square")? Could you discuss the existing literature on learning via observation (to name a few, Cooper et al., 2012; Liljeholm, Molloy, & O'Doherty, 2012; reviewed in Olsson, Knapska, & Lindström, 2020) in terms of single v. dual-process theory? I would appreciate more motivation and discussion on how the comparison between learning via observations vs. instructions about observations may parallel the comparison between learning via direct experiences vs. instructions discussed in Pages 5-7.

2. It would be informative for the readers if the DVs are motivated further in the introduction. What are the predictions for self-report rating measures and the behavioral measures (pIAT and SCR), given single-process propositional vs. dual-process theory?

3. In Experiments 1 and 2, the CS for evaluative responses was food, and preference for food could be quite subjective in nature. For the model's response to generalize to your own response to the presented food item, you would need to infer that you and the model share taste. Was there any debriefing question that could potentially address this? Perhaps participants' interpretation varied ("The model dislikes the cookie" vs "The cookie is bad"), and the repetition in the observation condition and the "disgust" reaction could have skewed participants' interpretation toward "the cookie is bad." Relatedly, what was the rationale behind showing the same video three times, as opposed to (1) the same model reacting to the same cookie on three different occasions, or (2) three different models reacting to the same cookie?

4. Experiment 1 Main Analyses: "The main effect of acquisition type was qualified by an interaction with block order [...], such that it was only significant if participants completed the learning-consistent block first." Could you provide statistics for each counterbalancing condition? What was the Bayes Factor for the condition where a learning inconsistent block was first (i.e., can you conclude that there is strong evidence for no difference between acquisition types, and if so, what is the interpretation/significance)?

5. In Experiments 3 and 4, on top of now looking at fear responses, the contingency has also changed to probabilistic learning. In my opinion, that is worth emphasizing. As the authors discuss in the discussion, probabilistic contingency may be better conveyed by instruction. Given that majority of participants in the two observation conditions believed that the contingency was "always," was there any difference in resilience to extinction in the test phase (trial x acquisition condition interaction in CS+)?

Minor points:

1. It would be useful to have methods figure that shows the experimental paradigm clearly.
2. For "BF10 >10000" there is a space between zeros "10 000"

Review form: Reviewer 3

Is the manuscript scientifically sound in its present form?

Yes

Are the interpretations and conclusions justified by the results?

Yes

Is the language acceptable?

Yes

Do you have any ethical concerns with this paper?

No

Have you any concerns about statistical analyses in this paper?

No

Recommendation?

Accept with minor revision (please list in comments)

Comments to the Author(s)

Comments to the Author:

Kasran and colleagues tested the effect of instructions about observations on learning/conditioning, as opposed to direct observation. Across a series of four experiments (Exp1-2 on evaluation on cookies; Exp 3-4 on fear learning) that build on top of one another, they observed clear evidence that instructions about observations could indeed shape learning experience, and reported mixed results regarding whether the effect of instructions is smaller or larger compared to actual observation.

These experiments are thematically and logically well connected and will have important implications to both basic research and applied/clinical research. Also they embraced as many open science practices as possible (pre-registration, open data, open code, transparency on deviating from original registration, Bayes Factor, etc), which make future replications easier and fits the scope of the journal very well.

This paper is generally well-written. I only have a main conceptual comment, and a few minor comments mainly for clarification.

Major point.

The authors observed mixed results between Exp1-2 and Exp3-4, and they indeed discussed the potential differences (P42). But I would like to emphasize that Exp1-2 and Exp 3-4 are essentially two different types of learning. In Exp1-2, the associations between CS and US are deterministic such that as long as the "model" sees the CS+, he will express a positive reaction. Watching a video is indeed more salient than merely reading instructions in text. In essence, in both actual vs instructed conditions, participants did not need to "figure out" anything on their own, instead, they copy what is shown via video vs text. And strength/intensity is likely to cause the difference here.

In Exp 3-4, however, the fear learning is probabilistic. In actual observations, participants did not know the CS-US association and they HAD TO LEARN them through observation. On the other hand, in the instruction condition, participants were clearly told the association such that they did not need to learn it. This is possibly why in Exp3, the effect of instruction group was slightly larger than the observation group. Essentially, if they were told what was the "truth" without having to "figure it out", they may learn better. That says, in actual observation, participants had

to figure out, yet in instructed observation, participants did not have to figure out. And only the latter is comparable to Exp1-2, and the former belongs to a slight distinctive class. I encourage the authors to discuss the different types of learning (see, Olsson et al 2020; Zhang and Glascher, 2020) in their context.

Very minor

- Grabenhorst et al (2019) might be a good recent reference for discussing observation learning in animals.
- In Exp1-2, in the instruction condition, did participants see a picture of the two logos?
- In Exp1-2, did the authors measure the evaluative ratings BEFORE the observations? This might be relevant because participants may have some a priori preference toward one of the cookies.
- For Exp1-2, although these results are straightforward, it would still be great to include a figure.

Reference:

Grabenhorst, F., Báez-Mendoza, R., Genest, W., Deco, G., & Schultz, W. (2019). Primate amygdala neurons simulate decision processes of social partners. *Cell*, 177(4), 986-998.

Olsson, A., Knapska, E., & Lindström, B. (2020). The neural and computational systems of social learning. *Nature Reviews Neuroscience*, 21(4), 197-212.

Zhang, L., & Gläscher, J. (2020). A brain network supporting social influences in human decision-making. *Science advances*, 6(34), eabb4159.

Decision letter (RSOS-211085.R0)

Dear Ms Kasran

The Editors assigned to your paper RSOS-211085 "Learning via instructions about observations: Exploring similarities and differences with learning via actual observations" have made a decision based on their reading of the paper and any comments received from reviewers.

Regrettably, in view of the reports received, the manuscript has been rejected in its current form. However, a new manuscript may be submitted which takes into consideration these comments.

We invite you to respond to the comments supplied below and prepare a resubmission of your manuscript. Below the referees' and Editors' comments (where applicable) we provide additional requirements. We provide guidance below to help you prepare your revision.

Please note that resubmitting your manuscript does not guarantee eventual acceptance, and we do not generally allow multiple rounds of revision and resubmission, so we urge you to make every effort to fully address all of the comments at this stage. If deemed necessary by the Editors, your manuscript will be sent back to one or more of the original reviewers for assessment. If the original reviewers are not available, we may invite new reviewers.

Please resubmit your revised manuscript and required files (see below) no later than 22-Feb-2022. Note: the ScholarOne system will 'lock' if resubmission is attempted on or after this deadline. If you do not think you will be able to meet this deadline, please contact the editorial office immediately.

Please note article processing charges apply to papers accepted for publication in Royal Society Open Science (<https://royalsocietypublishing.org/rsos/charges>). Charges will also apply to papers transferred to the journal from other Royal Society Publishing journals, as well as papers submitted as part of our collaboration with the Royal Society of Chemistry (<https://royalsocietypublishing.org/rsos/chemistry>). Fee waivers are available but must be requested when you submit your manuscript (<https://royalsocietypublishing.org/rsos/waivers>).

Thank you for submitting your manuscript to Royal Society Open Science and we look forward to receiving your resubmission. If you have any questions at all, please do not hesitate to get in touch.

on behalf of Dr Oliver Robinson (Associate Editor) and Essi Viding (Subject Editor)
openscience@royalsociety.org

Associate Editor Comments to Author (Dr Oliver Robinson):

Associate Editor: 1

Comments to the Author:

The reviewers found the paper interesting, but unfortunately identified some fairly major issues with the theoretical framing and underpinnings of the paper. Please see the reviews for specific comments. As a result we are rejecting the paper. However, we will allow resubmission of a revision if the authors believe they can thoroughly address these concerns.

Reviewer comments to Author:

Reviewer: 1

Comments to the Author(s)

In their manuscript "Learning via instructions about observations: Exploring similarities and differences with learning via actual observations" the authors report a set of experiments which documents the effect of "instructions about observations" on various affective learning indices. The basic finding is interesting - people can learn through hearing about other people's experiences - , but hardly surprising given how well documented various instructional learning effects are. Nonetheless, the experiments are carefully conducted and presented in a balanced manner. However, the theoretical framework is, in my view, weak, and in the end its not entirely clear what we have learned (except that people learn via hearing about others experiences). Rather than testing substantial hypotheses, the paper has a "its known that $1+1 = 2$, and $2+2 = 4$, but what about $1+2$?" - character. In other words, it documents a (expected) phenomenon. In my opinion, the paper would benefit from acknowledging this, rather than claiming to test theory-derived hypotheses.

My main issue is that the authors preferred "propositional perspective" is so vague that evaluating the logic of predictions/hypotheses is impossible. The main prediction seems to be "to the extent that actually observed events and instructions about those observed events result in the same or a similar proposition being formed and considered valid, this perspective predicts that both pathways would have a similar impact on behaviour." From this it's unclear both (i) what propositions are, and (ii) how the authors will measure whether similar propositions are formed.

Both issues could be addressed with a formal model that made it crystal clear what associations are, and make clear a priori assumptions about the validity of propositions.

Instead, the authors seem to find this level of unspecific reasoning sufficient. As consequence, it's very unclear what theoretical perspective the results support (as acknowledged in the discussion). The "propositional perspective" seems, due to its lack of specificity, nearly impossible to falsify. In the face of data that shows differences between different types of social information (which is not predicted by the propositional perspective by definition), the authors try to salvage it by a number of axillary hypotheses (pages 42-43).

For example, they propose that the number of pairings and the strength of reactions to stimuli could explain differences between "pathways". Why would a propositional mechanism care about such things? If learning is governed solely by the truth value of propositions, a single piece of information should be as efficient as multiple pieces. Or do the authors assume that truth value increases with repetitions? If so, the authors could consider using established and precise Bayesian learning models that implement exactly such belief updating and allows quantitative predictions.

What do other theories predict? The authors state "Based on a propositional perspective, we predicted that all three conditions would have a similar impact on evaluations of the cookies (with the CSpos being evaluated more positively than the CSneg)." But such statements are of course more meaningful if different theories make different apriori predictions.

For experiment 2, the authors state ""Based on a propositional perspective, we would predict that the effects in this enhanced-instructions condition would be (a) larger than the effects in the instructions condition and (b) similar in magnitude to the effects in the observations condition (given that the information they conveyed was now closer in nature)." Why would any additive effects be expected if propositions represent the truth value of relationships in the world?

In summary, these issues highlight the need for a formal specification (c.f. reinforcement learning and Bayesian learning) of the "propositional perspective" for it to be take serious as account of learning.

Minor.

- The concept of pathways is not well defined. Why would instructions about events befalling a third party represent a different pathway than instructions about events happening to oneself?

- The statistical models are sometimes rather mindlessly applied. Specifically, the authors report two ANOVA analyses with rather silly results (p 15): "The main effect of acquisition type was qualified by an interaction with stimulus assignment, $F(2, 260) = 3.51$, $p = .03$, such that the effect was only significant if Plogo served as the CSpos" and "The main effect of acquisition type was qualified by an interaction with block order, $F(2, 262) = 3.31$, $p = .038$, such that it was only significant if participants completed the learning-consistent block first." These results seem quite obviously spurious, and one wonder why these factors were included in the model. If taken seriously, the authors should discuss the limits to generalization.

- Was the US in Experiment 3 calibrated for each subject?

- Please specify the linear mixed-effects model (p 28)

- What evidence is there that participants in experiment 4 truthfully reported whether they wore headphones etc? In other words, why would participants be truthful in their reports? Did the

authors safeguard against participants thinking that they might lose monetary rewards if truthful?

Reviewer: 2

Comments to the Author(s)

In this paper, Kasran and colleagues investigate how getting instructions about another person's responses influences learning versus actually observing their responses. The authors report that both instructions about observation and actual observations are effective in influencing the learner's own evaluative (Experiments 1-2) and fear (Experiments 3-4) responses. Actual observations were more effective than instruction in some cases but not all.

This study provides valuable data to the field of social learning. As the authors point out in the introduction and discussion, the strength of behavioral change can vary depending on the pathways (in this case, actual observations vs instructions about observations) and the type of information being conveyed, and there are practical implications in finding out which ones are effective in which scenario. However, the theoretical implications are rather unclear to me. I will elaborate it below with some additional questions and comments.

1. What would be the theoretical implication for showing the effectiveness of instructions about observations (e.g., "A model would react negatively to an unpleasant sound that followed a yellow square") as opposed to instructions about the pairing itself (e.g., "An unpleasant sound will be played following a yellow square")? Could you discuss the existing literature on learning via observation (to name a few, Cooper et al., 2012; Liljeholm, Molloy, & O'Doherty, 2012; reviewed in Olsson, Knapska, & Lindström, 2020) in terms of single v. dual-process theory? I would appreciate more motivation and discussion on how the comparison between learning via observations vs. instructions about observations may parallel the comparison between learning via direct experiences vs. instructions discussed in Pages 5-7.

2. It would be informative for the readers if the DVs are motivated further in the introduction. What are the predictions for self-report rating measures and the behavioral measures (pIAT and SCR), given single-process propositional vs. dual-process theory?

3. In Experiments 1 and 2, the CS for evaluative responses was food, and preference for food could be quite subjective in nature. For the model's response to generalize to your own response to the presented food item, you would need to infer that you and the model share taste. Was there any debriefing question that could potentially address this? Perhaps participants' interpretation varied ("The model dislikes the cookie" vs "The cookie is bad"), and the repetition in the observation condition and the "disgust" reaction could have skewed participants' interpretation toward "the cookie is bad." Relatedly, what was the rationale behind showing the same video three times, as opposed to (1) the same model reacting to the same cookie on three different occasions, or (2) three different models reacting to the same cookie?

4. Experiment 1 Main Analyses: "The main effect of acquisition type was qualified by an interaction with block order [...], such that it was only significant if participants completed the learning-consistent block first." Could you provide statistics for each counterbalancing condition? What was the Bayes Factor for the condition where a learning inconsistent block was first (i.e., can you conclude that there is strong evidence for no difference between acquisition types, and if so, what is the interpretation/significance)?

5. In Experiments 3 and 4, on top of now looking at fear responses, the contingency has also changed to probabilistic learning. In my opinion, that is worth emphasizing. As the authors discuss in the discussion, probabilistic contingency may be better conveyed by instruction. given

that majority of participants in the two observation conditions believed that the contingency was "always," was there any difference in resilience to extinction in the test phase (trial x acquisition condition interaction in CS+)?

Minor points:

1. It would be useful to have methods figure that shows the experimental paradigm clearly.
2. For "BF10 >10000" there is a space between zeros "10 000"

Reviewer: 3

Comments to the Author(s)

Comments to the Author:

Kasran and colleagues tested the effect of instructions about observations on learning/conditioning, as opposed to direct observation. Across a series of four experiments (Exp1-2 on evaluation on cookies; Exp 3-4 on fear learning) that build on top of one another, they observed clear evidence that instructions about observations could indeed shape learning experience, and reported mixed results regarding whether the effect of instructions is smaller or larger compared to actual observation.

These experiments are thematically and logically well connected and will have important implications to both basic research and applied/clinical research. Also they embraced as many open science practices as possible (pre-registration, open data, open code, transparency on deviating from original registration, Bayes Factor, etc), which make future replications easier and fits the scope of the journal very well.

This paper is generally well-written. I only have a main conceptual comment, and a few minor comments mainly for clarification.

Major point.

The authors observed mixed results between Exp1-2 and Exp3-4, and they indeed discussed the potential differences (P42). But I would like to emphasize that Exp1-2 and Exp 3-4 are essentially two different types of learning. In Exp1-2, the associations between CS and US are deterministic such that as long as the "model" sees the CS+, he will express a positive reaction. Watching a video is indeed more salient than merely reading instructions in text. In essence, in both actual vs instructed conditions, participants did not need to "figure out" anything on their own, instead, they copy what is shown via video vs text. And strength/intensity is likely to cause the difference here.

In Exp 3-4, however, the fear learning is probabilistic. In actual observations, participants did not know the CS-US association and they HAD TO LEARN them through observation. On the other hand, in the instruction condition, participants were clearly told the association such that they did not need to learn it. This is possibly why in Exp3, the effect of instruction group was slightly larger than the observation group. Essentially, if they were told what was the "truth" without having to "figure it out", they may learn better. That says, in actual observation, participants had to figure out, yet in instructed observation, participants did not have to figure out. And only the latter is comparable to Exp1-2, and the former belongs to a slight distinctive class. I encourage the authors to discuss the different types of learning (see, Olsson et al 2020; Zhang and Glascher, 2020) in their context.

Very minor

- Grabenhorst et al (2019) might be a good recent reference for discussing observation learning in animals.
- In Exp1-2, in the instruction condition, did participants see a picture of the two logos?

- In Exp1-2, did the authors measure the evaluative ratings BEFORE the observations? This might be relevant because participants may have some a priori preference toward one of the cookies.
- For Exp1-2, although these results are straightforward, it would still be great to include a figure.

Reference:

Grabenhorst, F., Báez-Mendoza, R., Genest, W., Deco, G., & Schultz, W. (2019). Primate amygdala neurons simulate decision processes of social partners. *Cell*, 177(4), 986-998.

Olsson, A., Knapska, E., & Lindström, B. (2020). The neural and computational systems of social learning. *Nature Reviews Neuroscience*, 21(4), 197-212.

Zhang, L., & Gläscher, J. (2020). A brain network supporting social influences in human decision-making. *Science advances*, 6(34), eabb4159.

===PREPARING YOUR MANUSCRIPT===

===PREPARING YOUR REVISION IN SCHOLARONE===

Author's Response to Decision Letter for (RSOS-211085.R0)

See Appendix A.

RSOS-220059.R0

Review form: Reviewer 1

Is the manuscript scientifically sound in its present form?

Yes

Are the interpretations and conclusions justified by the results?

Yes

Is the language acceptable?

Yes

Do you have any ethical concerns with this paper?

No

Have you any concerns about statistical analyses in this paper?

Yes

Recommendation?

Accept with minor revision (please list in comments)

Comments to the Author(s)

In their revision, the authors have generally clarified the issues I was concerned with. The phenomenon is of interest, and the authors' presentation is nuanced.

On the more general point: the authors have provided more information about the propositional perspective. Based on this information, I really wonder what the difference to model-based reinforcement learning is? In model-based RL, the model typically includes both information about the value and identity of the outcome, and an estimate of the probability that the outcome will transpire. As such, such a model would specify whether the CS “sometimes predict” the US, and the identity of the US. It would be helpful to acknowledge this similarity.

Although I appreciate that a full theoretical exegesis is beyond the scope of the present paper, I urge the authors to, in future work, integrate their propositional account with modern, computational accounts of learning, rather than rely on verbal contrast against various (strawman, in my opinion) “associative” accounts of learning.

Minor: the random effects structure used in the linear mixed models (random intercept by subject) is known to be too liberal in many cases (see <https://www.ncbi.nlm.nih.gov/pmc/articles/PMC3881361/> & <https://www.ncbi.nlm.nih.gov/pmc/articles/PMC2657178/>). Given that the authors are analyzing repeated measures data, the correct (in terms of error control) random effects structure usually includes random slopes for all fixed effects (including interactions).

Review form: Reviewer 2

Is the manuscript scientifically sound in its present form?

Yes

Are the interpretations and conclusions justified by the results?

Yes

Is the language acceptable?

Yes

Do you have any ethical concerns with this paper?

No

Have you any concerns about statistical analyses in this paper?

No

Recommendation?

Accept with minor revision (please list in comments)

Comments to the Author(s)

The revised manuscript by Kasran and colleagues includes more detailed theoretical discussions, and I found the authors' response to the comments to be addressing most of major points. However, I'd like to further clarify a few points.

I appreciate the authors' clarification on the differences between learning via instructions and learning via instructions about observations, where informational contents differ. Related to this point and the theoretical implications, the key prediction under the propositional perspective seems to be that, when the information contents are matched (or, more precisely, if the learned information is matched), there should be no difference in learning, and the potential source of learning difference between observations and instructions about observations can be attributed to the difference in information. To this end, authors have taken measures to match informational contents (Exp 2 and Exp 4), and interpret the remaining differences in learning under this perspective.

However, I am still slightly confused about the rationale for repeating the same recorded videos. If participants interpret the repetition as "this model tasted the cookie once and I'm watching it three times," participants' own ratings of CS should not change as a function of the number of repetitions. However, if they interpret the repetition as "this model tasted the cookie three times and ALWAYS showed the same reaction," the number of repetitions should strengthen the beliefs about the relationship between CS and UR. Even though consensus is beyond the scope of this paper (I am excited to hear that the authors are exploring the link to attribution in the lab!), I think it would be worth discussing whether participants beliefs about the stimuli in the observation condition would lead to different predictions under associative vs propositional theories. This point is less significant in Experiment 3 and 4, where participants were told that the model went through the pairings multiple times (and due to multiple types of video clips, it may have been less obvious that the same recordings were repeating), but I think it would be worth asking the participants about their interpretations of repeated recordings in future studies, in light of informational content and believability that authors discuss.

Minor comment:

In Figures 1 and 4, it would be helpful to clarify in the caption that the videos that were shown to participants did not have the faces masked with labels. It may be more informative if these were replaced with illustrations, if there is a need to not reveal the stimuli in the paper.

Review form: Reviewer 3

Is the manuscript scientifically sound in its present form?

Yes

Are the interpretations and conclusions justified by the results?

Yes

Is the language acceptable?

Yes

Do you have any ethical concerns with this paper?

No

Have you any concerns about statistical analyses in this paper?

No

Recommendation?

Accept with minor revision (please list in comments)

Comments to the Author(s)

The authors did a great job in revising their manuscript. It is much clearer and the conceptual differences between their experiments were also made explicit and discussed in detail.

Only a small thing: one of the references I mentioned in my previous comment (Zhang & Gläscher, 2020) was not cited in the revised manuscript. And I'd also encourage the authors to mention a very recent work on observational fear learning (Haaker et al., 2021). Otherwise, I have no other comments, and congrats on this work.

Reference:

Haaker, J., Diaz-Mataix, L., Guillazo-Blanch, G., Stark, S. A., Kern, L., LeDoux, J. E., & Olsson, A. (2021). Observation of others' threat reactions recovers memories previously shaped by firsthand experiences. *Proceedings of the National Academy of Sciences*, 118(30).

Zhang, L., & Gläscher, J. (2020). A brain network supporting social influences in human decision-making. *Science advances*, 6(34), eabb4159.

Decision letter (RSOS-220059.R0)

Dear Ms Kasran

On behalf of the Editors, we are pleased to inform you that your Manuscript RSOS-220059 "Learning via instructions about observations: Exploring similarities and differences with learning via actual observations" has been accepted for publication in Royal Society Open Science subject to minor revision in accordance with the referees' reports. Please find the referees' comments along with any feedback from the Editors below my signature.

Please submit your revised manuscript and required files (see below) no later than 7 days from today's (ie 22-Feb-2022) date. Note: the ScholarOne system will 'lock' if submission of the revision is attempted 7 or more days after the deadline. If you do not think you will be able to meet this deadline please contact the editorial office immediately.

on behalf of Dr Oliver Robinson (Associate Editor) and Essi Viding (Subject Editor)
openscience@royalsociety.org

Associate Editor Comments to Author (Dr Oliver Robinson):

Associate Editor

Comments to the Author:

Thank you for a comprehensive revision. We are delighted to accept your paper pending the final small tweaks suggested by the reviewers.

Reviewer comments to Author:

Reviewer: 3

Comments to the Author(s)

The authors did a great job in revising their manuscript. It is much clearer and the conceptual differences between their experiments were also made explicit and discussed in detail.

Only a small thing: one of the references I mentioned in my previous comment (Zhang & Gläscher, 2020) was not cited in the revised manuscript. And I'd also encourage the authors to mention a very recent work on observational fear learning (Haaker et al., 2021). Otherwise, I have no other comments, and congrats on this work.

Reference:

Haaker, J., Diaz-Mataix, L., Guillazo-Blanch, G., Stark, S. A., Kern, L., LeDoux, J. E., & Olsson, A. (2021). Observation of others' threat reactions recovers memories previously shaped by firsthand experiences. *Proceedings of the National Academy of Sciences*, 118(30).

Zhang, L., & Gläscher, J. (2020). A brain network supporting social influences in human decision-making. *Science advances*, 6(34), eabb4159.

Reviewer: 2

Comments to the Author(s)

The revised manuscript by Kasran and colleagues includes more detailed theoretical discussions, and I found the authors' response to the comments to be addressing most of major points. However, I'd like to further clarify a few points.

I appreciate the authors' clarification on the differences between learning via instructions and learning via instructions about observations, where informational contents differ. Related to this point and the theoretical implications, the key prediction under the propositional perspective seems to be that, when the information contents are matched (or, more precisely, if the learned information is matched), there should be no difference in learning, and the potential source of learning difference between observations and instructions about observations can be attributed to the difference in information. To this end, authors have taken measures to match informational contents (Exp 2 and Exp 4), and interpret the remaining differences in learning under this perspective.

However, I am still slightly confused about the rationale for repeating the same recorded videos. If participants interpret the repetition as "this model tasted the cookie once and I'm watching it three times," participants' own ratings of CS should not change as a function of the number of repetitions. However, if they interpret the repetition as "this model tasted the cookie three times and ALWAYS showed the same reaction," the number of repetitions should strengthen the beliefs about the relationship between CS and UR. Even though consensus is beyond the scope of this paper (I am excited to hear that the authors are exploring the link to attribution in the lab!), I think it would be worth discussing whether participants beliefs about the stimuli in the observation condition would lead to different predictions under associative vs propositional theories. This point is less significant in Experiment 3 and 4, where participants were told that the model went through the pairings multiple times (and due to multiple types of video clips, it may have been less obvious that the same recordings were repeating), but I think it would be worth asking the participants about their interpretations of repeated recordings in future studies, in light of informational content and believability that authors discuss.

Minor comment:

In Figures 1 and 4, it would be helpful to clarify in the caption that the videos that were shown to participants did not have the faces masked with labels. It may be more informative if these were replaced with illustrations, if there is a need to not reveal the stimuli in the paper.

Reviewer: 1

Comments to the Author(s)

In their revision, the authors have generally clarified the issues I was concerned with. The phenomenon is of interest, and the authors' presentation is nuanced.

On the more general point: the authors have provided more information about the propositional perspective. Based on this information, I really wonder what the difference to model-based reinforcement learning is? In model-based RL, the model typically includes both information about the value and identity of the outcome, and an estimate of the probability that the outcome will transpire. As such, such a model would specify whether the CS "sometimes predict" the US, and the identity of the US. It would be helpful to acknowledge this similarity.

Although I appreciate that a full theoretical exegesis is beyond the scope of the present paper, I urge the authors to, in future work, integrate their propositional account with modern, computational accounts of learning, rather than rely on verbal contrast against various (strawman, in my opinion) "associative" accounts of learning.

Minor: the random effects structure used in the linear mixed models (random intercept by subject) is known to be too liberal in many cases (see <https://www.ncbi.nlm.nih.gov/pmc/articles/PMC3881361/> & <https://www.ncbi.nlm.nih.gov/pmc/articles/PMC2657178/>). Given that the authors are analyzing repeated measures data, the correct (in terms of error control) random effects structure usually includes random slopes for all fixed effects (including interactions).

===PREPARING YOUR MANUSCRIPT===

one version should clearly identify all the changes that have been made (for instance, in coloured highlight, in bold text, or tracked changes);

===PREPARING YOUR REVISION IN SCHOLARONE===

-- If you are requesting an article processing charge waiver, you must select the relevant waiver option (if requesting a discretionary waiver, the form should have been uploaded, see 'File upload' above).

-- If you have uploaded any electronic supplementary (ESM) files, please ensure you follow the guidance at <https://royalsociety.org/journals/authors/author-guidelines/#supplementary-material> to include a suitable title and informative caption. An example of appropriate titling and captioning may be found at https://figshare.com/articles/Table_S2_from_Is_there_a_trade-off_between_peak_performance_and_performance_breadth_across_temperatures_for_aerobic_scope_in_teleost_fishes_/3843624.

Author's Response to Decision Letter for (RSOS-220059.R0)

See Appendix B.

Decision letter (RSOS-220059.R1)

Dear Ms Kasran,

I am pleased to inform you that your manuscript entitled "Learning via instructions about observations: Exploring similarities and differences with learning via actual observations" is now accepted for publication in Royal Society Open Science.

on behalf of Dr Oliver Robinson (Associate Editor) and Essi Viding (Subject Editor)
openscience@royalsociety.org

Appendix A

Response to Decision Letter

Manuscript Title: Learning via instructions about observations: Exploring similarities and differences with learning via actual observations

Manuscript ID: RSOS-211085

Associate Editor Comments to Author (Dr Oliver Robinson)

Associate Editor Comment #1:

The reviewers found the paper interesting, but unfortunately identified some fairly major issues with the theoretical framing and underpinnings of the paper. Please see the reviews for specific comments. As a result we are rejecting the paper. However, we will allow resubmission of a revision if the authors believe they can thoroughly address these concerns.

Authors: We thank the Editor and Reviewers 1-3 for their feedback. We have revised the paper in line with this feedback. Most importantly, we have (a) clarified the role of the propositional perspective in our research, (b) included a more elaborate and clear description of the propositional perspective, (c) added a section discussing the current comparison between instructions about observations and observations as well as the theories this comparison may serve to inform, (d) motivated the dependent variables in light of predictions of different theoretical perspectives, and (e) added a discussion of the possible implications of the contingency being deterministic in Experiments 1-2 but probabilistic in Experiments 3-4. Please see below for detailed responses to the Reviewers' comments (please note that page numbers refer to the version with tracked changes).

Reviewer Comments to Author

Reviewer: 1

Reviewer 1 Comment #1:

In their manuscript "Learning via instructions about observations: Exploring similarities and differences with learning via actual observations" the authors report a set of experiments which documents the effect of "instructions about observations" on various affective learning indices. The basic finding is interesting – people can learn through hearing about other people's experiences -, but hardly surprising given how well documented various instructional learning effects are. Nonetheless, the experiments are carefully conducted and presented in a balanced manner. However, the theoretical framework is, in my view, weak, and in the end its not entirely clear what we have learned (except that people learn via hearing about others experiences). Rather than testing substantial hypotheses, the paper has a "its known that $1+1 = 2$, and $2+2 = 4$, but what about $1+2$?" – character. In other words, it documents a (expected) phenomenon. In my opinion, the paper would benefit from acknowledging this, rather than claiming to test theory-derived hypotheses.

My main issue is that the authors preferred "propositional perspective" is so vague that evaluating the logic of predictions/hypotheses is impossible. The main prediction seems to be "to the extent that actually observed events and instructions about those observed events result in the same or a similar proposition being formed and considered valid, this perspective predicts that both pathways would have a similar impact on behaviour." From this it's unclear both (i) what propositions are, and (ii) how the authors will measure whether similar propositions are formed. Both issues could be addressed with a formal model that made it crystal clear what associations are, and make clear a priori assumptions about the validity of propositions.

Instead, the authors seem to find this level of unspecific reasoning sufficient. As consequence, it's very unclear what theoretical perspective the results support (as acknowledged in the discussion). The "propositional perspective" seems, due to its lack of specificity, nearly impossible to falsify. In the face of data that shows differences between different types of social information (which is not predicted by the propositional perspective by definition), the authors try to salvage it by a number of axillary hypotheses (pages 42-43).

For example, they propose that the number of pairings and the strength of reactions to stimuli could explain differences between "pathways". Why would a propositional mechanism care about such things? If learning is governed solely by the truth value of propositions, a single piece of information should be as efficient as multiple pieces. Or do the authors assume that truth value increases with repetitions? If so, the authors could consider using established and precise Bayesian learning models that implement exactly such belief updating and allows quantitative predictions.

What do other theories predict? The authors state "Based on a propositional perspective, we predicted that all three conditions would have a similar impact on evaluations of the cookies (with the CSpos being evaluated more positively than the CSneg)." But such statements are of course more meaningful if different theories make different apriori predictions.

For experiment 2, the authors state ""Based on a propositional perspective, we would predict that the effects in this enhanced-instructions condition would be (a) larger than the effects in the instructions condition and (b) similar in magnitude to the effects in the observations condition (given that the information they conveyed was now closer in nature)." Why would any additive effects be expected if propositions represent the truth value of relationships in the world?

In summary, these issues highlight the need for a formal specification (c.f. reinforcement learning and Bayesian learning) of the "propositional perspective" for it to be take serious as account of learning.

Authors: Reviewer 1 makes three main points here, which we address separately below.

Point 1: Need for formalization of the propositional perspective

The Reviewer's first point is that our manuscript documents a phenomenon rather than tests theory-derived hypotheses that would allow us to falsify the propositional perspective. Relatedly, the Reviewer notes that the propositional perspective could probably not be falsified in any case, because it is too vague and unspecific. The suggested strategy is to develop the perspective into a formal model that could be used to derive highly specific predictions that allow for falsification.

On the one hand, we acknowledge that we could have been more precise in our outline of the propositional perspective in the Introduction. Towards this end, we have revised the manuscript to better outline the perspective (see *Point 2* below).

On the other hand, we politely disagree with the point that the paper should be revised in order to formalize the propositional perspective. This point reflects a view on science (focusing strongly on formalization and falsification) that is different from our view. Arguably, if falsification is seen as crucial in science, then few findings in psychology would deserve publication because data rarely allow for the falsification of theories. Even when theories are formalized, they typically have a protective belt due to possible variations in parameters, and even models that have been falsified (e.g., the Rescorla-Wagner model) continue to be used. Therefore, although we certainly agree that there is merit in formal and falsifiable theories, we believe that there is also merit in broad perspectives that have substantial orienting and predictive value (i.e., that help us to highlight new, unexplored topics and make novel predictions).

The propositional perspective falls within this latter category. It is true that it has not been formalized and is difficult (if not impossible) to falsify (see De Houwer, 2018). However, it does have utility: it can help to orientate researchers toward phenomena that were not previously documented or studied and to generate new empirical findings that have practical implications and inform theoretical thinking (i.e., that constrain theories; see De Houwer, 2018; De Houwer et al., 2021). In other words, even if the resulting findings cannot *falsify* the propositional perspective or other available perspectives, there is merit in documenting these new phenomena and their moderators.

In line with this meta-science view, we would also like to clarify that the intent of our discussion on p. 45-48 was not to salvage the propositional perspective, but rather to use this perspective to reflect on how one might explain differences between pathways, thereby generating novel predictions about the sources of those differences. Testing these predictions could lead to the discovery of empirical knowledge on learning effects, irrespective of whether the predictions are supported or not.

In sum, rather than leave out the material on the available theoretical perspectives altogether or attempt to arrive at a formalization of the propositional perspective (which was not the goal of the current research), we have decided to clarify in the Introduction our view on the propositional perspective and its role in the current research (see p. 12).

Point 2: Parameters of the propositional perspective

The Reviewer asks several questions about why certain things should matter from a propositional perspective. These questions seem to arise from an interpretation of the propositional perspective that focuses solely on the truth value of propositions and not on their content.

We have therefore revised our manuscript to provide a more elaborate explanation of the propositional perspective in the Introduction (see p. 6-7). Most importantly, we now discuss how propositions are defined in terms of their informational content: they can specify the nature of the relation between events. For example, the proposition that “the blue square sometimes predicts a mild electric shock” is not only truth verifiable but also contains information about the events (“blue square”, “mild electric shock”) and the specific relation between them (“sometimes predicts”). This hopefully provides an answer to the Reviewer’s various questions:

1. The Reviewer asked why the *number of pairings* should matter from a propositional perspective. This question perhaps refers to what we say in the Discussion, namely that the *assumed strength of the contingency* should play a role in the obtained effects. This point was about the relational component of propositions. For example, if instructions about observations led most participants to form the proposition that “the blue square *sometimes* predicts an unpleasant sound” while actual observations led most participants to form the proposition that “the blue square *always* predicts an unpleasant sound” (as seemed to be the case based on the exploratory questions in Experiment 4), then this can explain why the latter group showed stronger fear responding.
2. Assumptions about the precise nature of the individual events also matter from a propositional perspective. For example, if actually seeing the model reactions led participants to assume that a cookie induced a *highly negative reaction*, whereas mere instructions led them to assume that the cookie induced a *slightly negative reaction*, we would expect a larger change in liking in the first condition. We hope that this clarifies (1) why the assumed strength of modelled reactions would matter based on a propositional perspective, and (2) why including examples of the reactions in the instructions was predicted to make the effects more similar to those of observations in Experiment 2.

Point 3: Predictions of other perspectives

Third, the Reviewer asks what other theories would predict and whether these predictions differ from our predictions. Although it can be challenging to derive clear predictions from some of the available perspectives, we have elaborated upon the predictions of dual-process theories to broadly convey the point that they would predict that instructions should have weaker effects on automatic evaluations and physiological fear responses (see p. 7-8). We hope that these additions make it clear that not all available theories predict that all behavioural changes should be similar across pathways.

Minor.

Reviewer 1 Comment #2:

- The concept of pathways is not well defined. Why would instructions about events befalling a third party represent a different pathway than instructions about events happening to oneself?

Authors: We have now restricted the use of the concept “pathway” to the traditional three learning pathways mentioned in the learning literature (e.g., Rachman, 1977): learning via direct experience, learning via observation, and learning via instructions (regardless of whether the instructions are about direct experiences or observations). As such, the paper no longer implies that instructions about observations constitute a different pathway than other types of instructions.

Reviewer 1 Comment #3:

- The statistical models are sometimes rather mindlessly applied. Specifically, the authors report two ANOVA analyses with rather silly results (p 15): “The main effect of acquisition type was qualified by an interaction with stimulus assignment, $F(2, 260) = 3.51, p = .03$, such that the effect was only significant if Plogo served as the CSpos” and “The main effect of acquisition type was qualified by an interaction with block order, $F(2, 262) = 3.31, p = .038$, such that it was only significant if participants completed the learning-consistent block first.” These results seem quite obviously spurious, and one wonders why these factors were included in the model. If taken seriously, the authors should discuss the limits to generalization.

Authors: We pre-registered that we would test the main effect of acquisition type again when the potential impact of other method factors was taken into account. Whereas the main effect remained significant in both of the cases above, we were hesitant to report this without mentioning the presence of statistically significant interactions with this main effect. However, we agree that these particular interactions are likely to be spurious findings and do not pose a threat to generalization. Therefore, we have clarified our intent in the Analytic Strategy, moved these results to the footnotes, included Bayes Factors for these interactions, and indicated that the results are reported for the sake of completeness but are likely spurious based on the Bayes Factors (see p. 18-20).

Reviewer 1 Comment #4:

- Was the US in Experiment 3 calibrated for each subject?

Authors: In line with the observational fear conditioning protocol (which uses electric shocks), we did not calibrate the sound for each participant (Haaker et al., 2017). This was not necessary because participants were never actually exposed to the sound themselves (i.e., all learning was based purely on the model’s observed behaviour). We have revised the paper to clarify this point (see p. 30).

Reviewer 1 Comment #5:

- Please specify the linear mixed-effects model (p 28)

Authors: We have specified which fixed and random effects were included in the model (see p. 33).

Reviewer 1 Comment #6:

- What evidence is there that participants in experiment 4 truthfully reported whether they wore headphones etc? In other words, why would participants be truthful in their reports? Did the authors safeguard against participants thinking that they might lose monetary rewards if truthful?

Authors: We acknowledge that we cannot guarantee that all participants responded truthfully to this question. However, within this question we did explicitly emphasize that their answer would not affect the payment they would receive for their participation and that it was very important that they answered truthfully. We have now clarified this in the paper (see p. 39).

Reviewer: 2

Reviewer 2 Comment #1:

In this paper, Kasran and colleagues investigate how getting instructions about another person's responses influences learning versus actually observing their responses. The authors report that both instructions about observation and actual observations are effective in influencing the learner's own evaluative (Experiments 1-2) and fear (Experiments 3-4) responses. Actual observations were more effective than instruction in some cases but not all.

This study provides valuable data to the field of social learning. As the authors point out in the introduction and discussion, the strength of behavioral change can vary depending on the pathways (in this case, actual observations vs instructions about observations) and the type of information being conveyed, and there are practical implications in finding out which ones are effective in which scenario. However, the theoretical implications are rather unclear to me. I will elaborate it below with some additional questions and comments.

1. What would be the theoretical implication for showing the effectiveness of instructions about observations (e.g., "A model would react negatively to an unpleasant sound that followed a yellow square") as opposed to instructions about the pairing itself (e.g., "An unpleasant sound will be played following a yellow square")? Could you discuss the existing literature on learning via observation (to name a few, Cooper et al., 2012; Liljeholm, Molloy, & O'Doherty, 2012; reviewed in Olsson, Knapska, & Lindström, 2020) in terms of single v. dual-process theory? I would appreciate more motivation and discussion on how the comparison between learning via observations vs. instructions about observations may parallel the comparison between learning via direct experiences vs. instructions discussed in Pages 5-7.

Authors: The Reviewer makes three points which we respond to below.

First, the Reviewer asks what the theoretical implication would be of showing that instructions about observations are effective given that we already know that instructions about pairings are. It is true that merely demonstrating that instructions about observations are effective does not have strong theoretical implications on top of demonstrating that people can learn via instructions about direct experience. Nevertheless, learning via instructions about observation is a novel phenomenon that had not yet been studied empirically. The information provided by instructions about observations clearly differs from the information provided by instructions about direct experiences.

In this regard, we should clarify that the instructions in our studies were more limited than what the Reviewer states here: we did not say that the model would react negatively to *an unpleasant sound that followed a yellow square* (which does provide direct information about the relevant square-sound contingency) but rather that the model would sometimes react negatively after a yellow

square disappeared from screen (which requires participants to infer the presence of the sound from the model's reactions, just like they would need to if they observed the actual reactions). Although this difference may seem subtle, it underlines the indirectness of instructions about observations. In a way, these constitute an indirect pathway (instructions) describing what can be experienced via another indirect pathway (observations). The fact that instructions about observations lead to clear behavioural changes therefore further highlights the remarkable capacity of humans to learn via indirect pathways. We now discuss this point in the paper (see p. 9).

Although the unique theoretical implications may not be extensive, we do believe that the current comparison between instructions about observations and actual observations can *inform* theorizing (as a side note, we have changed the title of the relevant section in the Discussion from "Theoretical Implications" to "Theoretical Considerations" to more clearly reflect this).

Specifically, the current comparison may serve to inform theories that include assumptions about the processes mediating observational learning. In line with the Reviewer's second request, we therefore briefly discuss single vs. dual-process theories of observational learning on p. 9-10. Note, however, that given the size of the literature on observational learning, we believe that a more detailed discussion would go beyond the scope of the current paper. Although most dual-process theories would likely still be able to explain instructions about observations having highly similar effects as actual observations, such a finding would be more in line with the idea that learning via instructions and learning via observations are both mediated by the same propositional process. While there have been a few studies that included a comparison between learning via observations and learning via instructions (Olsson & Phelps, 2004; Lindström et al., 2019), the informational content always differed between the two pathways (i.e., the instructions described future direct experiences rather than a model's reactions to stimuli) and any discrepancies could therefore have been due to a difference in *content* rather than a difference in *process*. Here, we conducted a direct comparison where the informational content was the same.

This also answers the Reviewer's third question: the current comparison indeed parallels the comparison between direct experiences and instructions about direct experiences. Specifically, the latter comparison also compares pathways while closely matching the conveyed information (e.g., reading "The blue square will be followed by a shock on 4 out of 6 trials" vs. experiencing a shock after the blue square on 4 out of 6 trials). We did not necessarily expect our results to deviate from prior results obtained based on this latter comparison, which is why we used this literature to guide our own research.

Because our results were quite mixed, their theoretical implications are – as the Reviewer notes – not clear. Nevertheless, we do document an intriguing new phenomenon. Moreover, our results stimulate reflection on possible sources of differences in order to inform future theorizing and inspire future research, which is what we tried to do in the Discussion.

We have added a new section to the paper which covers the above points (see p. 9-11).

Reviewer 2 Comment #2:

2. *It would be informative for the readers if the DVs are motivated further in the introduction. What are the predictions for self-report rating measures and the behavioral measures (pIAT and SCR), given single-process propositional vs. dual-process theory?*

Authors: We now discuss the predictions of the different theories in more detail in the Introduction (see p. 7-8). Although it can be challenging to derive clear predictions from the available dual-process perspectives, they usually differ most from the propositional perspective with regard to predictions for more automatic measures (in the sense that they often predict instructions to have weaker

effects on those measures than pairings). We refer back to this point in order to motivate our inclusion of more automatic measures (in addition to self-reports) in the Current Research section (see p. 11).

Reviewer 2 Comment #3:

3. In Experiments 1 and 2, the CS for evaluative responses was food, and preference for food could be quite subjective in nature. For the model's response to generalize to your own response to the presented food item, you would need to infer that you and the model share taste. Was there any debriefing question that could potentially address this? Perhaps participants' interpretation varied ("The model dislikes the cookie" vs "The cookie is bad"), and the repetition in the observation condition and the "disgust" reaction could have skewed participants' interpretation toward "the cookie is bad." Relatedly, what was the rationale behind showing the same video three times, as opposed to (1) the same model reacting to the same cookie on three different occasions, or (2) three different models reacting to the same cookie?

Authors: Good point. It is indeed plausible that participants' beliefs about whether they share taste preferences with the model would matter (at least from a propositional perspective). However, such a belief would seem to be relevant regardless of whether participants are informed about the model's reactions via instructions or via observations. Honestly, because we did not expect this belief to vary between the conditions in our studies, we did not include a debriefing question about this. To our knowledge, there is no theoretical reason why repetition might have skewed participants' interpretation toward "the cookie is bad". Nevertheless, we do see the value of including such an exploratory question in future research with this paradigm as it may indeed provide more insight into participants' responses. We now mention this in the Limitations and Future Directions section of the manuscript (see p. 50).

To answer the Reviewer's second question, the rationale behind showing the same videos multiple times was that we wanted to expose participants to repeated pairings of the CSs and the model's reactions without providing any additional information that was not included in the instructions condition. Specifically, showing the same model reacting to the cookies several times would have provided information about the consistency of his reactions, while showing three different models reacting to the cookies would have provided information about the consensus across different people. Attribution research (e.g., Kelley & Michela, 1980) has shown that consistency and consensus can both increase the likelihood that a given behaviour is attributed to the stimulus (rather than the person's general disposition or the specific circumstances, for example). In other words, such information could have increased the likelihood that participants attributed the reactions to the cookies. In the absence of consensus or consistency information in the instructions condition, showing multiple different videos would have created an important difference between conditions that was not relevant to our research questions.

We agree that these considerations raise interesting new questions for research. In fact, we have already explored the link between the attribution literature and learning via observation in recent studies conducted at our lab. We hope to report these studies in future papers.

Reviewer 2 Comment #4:

4. Experiment 1 Main Analyses: "The main effect of acquisition type was qualified by an interaction with block order [...], such that it was only significant if participants completed the learning-consistent block first." Could you provide statistics for each counterbalancing condition? What was the Bayes Factor for the condition where a learning inconsistent block was first (i.e., can you conclude that there is strong evidence for no difference between acquisition types, and if so, what is the interpretation/significance)?

Authors: For the condition where the learning consistent block was completed first, the statistics for the main effect of acquisition type are $F(2, 129) = 6.95, p = .001, BF_{10} = 22.71$; for the condition where the learning-inconsistent block was completed first, the statistics are $F(2, 133) = 2.46, p = .090, BF_{10} = 0.56$. In other words, there is no clear evidence for the absence of differences between acquisition types in the latter counterbalancing condition. We also agree with Reviewer 1 (see Reviewer 1 Comment #3) that the interaction is likely spurious (especially since the BF was also small, $BF_{10} = 1.23$). As our main goal was to check if the main effect of acquisition type remained significant when these method factors were taken into account, we now simply report in the text that the main effect remained significant and report the interaction in the footnotes and Supplementary Materials just for the sake of completeness, while noting that it is likely a spurious finding (see p. 19-20).

Reviewer 2 Comment #5:

5. In Experiments 3 and 4, on top of now looking at fear responses, the contingency has also changed to probabilistic learning. In my opinion, that is worth emphasizing. As the authors discuss in the discussion, probabilistic contingency may be better conveyed by instruction.

Authors: We thank the Reviewer for drawing our attention to this important difference. We now mention this explicitly in the introduction to Experiment 3 (see p. 26). In response to Reviewer 3 Comment #1, we also return to this point in more detail in the Discussion (see p. 47).

Reviewer 2 Comment #6:

given that majority of participants in the two observation conditions believed that the contingency was "always," was there any difference in resilience to extinction in the test phase (trial x acquisition condition interaction in CS+)?

Authors: Visually, the drop in ratings (especially expectancy ratings) after the first CS+ trial indeed seemed to be slightly more pronounced in the observations conditions, where most participants believed that the CS+ was always followed by the sound. However, this was not supported by the analyses. The extinction rate did not differ as a function of acquisition type, as indicated by the non-significant trial x acquisition interaction (fear: $F(5.53, 517.41) = 1.49, p = .19, \eta^2_p = 0.02, [0.00, 0.03]$; expectancy: $F(5.34, 499.01) = 1.46, p = .20, \eta^2_p = 0.02, 90\% \text{ CI } [0.00, 0.03]$). The conclusion is the same if we look separately at the ratings of the CS+ (respectively, $p = .61$ and $p = .15$). Of course, statistical power was likely too low to detect an interaction effect that we might expect to be quite small (as a substantial number of participants in the observations condition did believe that the CS+ was sometimes followed by the sound and a substantial number in the instructions condition believed that the CS+ was always followed by the sound).

Minor points:

Reviewer 2 Comment #7:

1. It would be useful to have methods figure that shows the experimental paradigm clearly.

Authors: We have included figures depicting the different acquisition types in Experiments 1-2 (Figure 1) and Experiments 3-4 (Figure 4).

Reviewer 2 Comment #8:

2. For "BF10 >10000" there is a space between zeros "10 000"

Authors: We have removed the space between zeros for all instances of $BF_{10} > 10000$.

Reviewer: 3

Reviewer 3 Comment #1:

Kasran and colleagues tested the effect of instructions about observations on learning/conditioning, as opposed to direct observation. Across a series of four experiments (Exp1-2 on evaluation on cookies; Exp 3-4 on fear learning) that build on top of one another, they observed clear evidence that instructions about observations could indeed shape learning experience, and reported mixed results regarding whether the effect of instructions is smaller or larger compared to actual observation. These experiments are thematically and logically well connected and will have important implications to both basic research and applied/clinical research. Also they embraced as many open science practices as possible (pre-registration, open data, open code, transparency on deviating from original registration, Bayes Factor, etc), which make future replications easier and fits the scope of the journal very well.

This paper is generally well-written. I only have a main conceptual comment, and a few minor comments mainly for clarification.

Major point.

The authors observed mixed results between Exp1-2 and Exp3-4, and they indeed discussed the potential differences (P42). But I would like to emphasize that Exp1-2 and Exp 3-4 are essentially two different types of learning. In Exp1-2, the associations between CS and US are deterministic such that as long as the “model” sees the CS+, he will express a positive reaction. Watching a video is indeed more salient than merely reading instructions in text. In essence, in both actual vs instructed conditions, participants did not need to “figure out” anything on their own, instead, they copy what is shown via video vs text. And strength/intensity is likely to cause the difference here.

In Exp 3-4, however, the fear learning is probabilistic. In actual observations, participants did not know the CS-US association and they HAD TO LEARN them through observation. On the other hand, in the instruction condition, participants were clearly told the association such that they did not need to learn it. This is possibly why in Exp3, the effect of instruction group was slightly larger than the observation group. Essentially, if they were told what was the “truth” without having to “figure it out”, they may learn better. That says, in actual observation, participants had to figure out, yet in instructed observation, participants did not have to figure out. And only the latter is comparable to Exp1-2, and the former belongs to a slight distinctive class. I encourage the authors to discuss the different types of learning (see, Olsson et al 2020; Zhang and Glascher, 2020) in their context.

Authors: We thank the Reviewer for drawing our attention to this important difference between experiments. We now mention in the introduction to Experiment 3 that the regularity is probabilistic in nature (see p. 26), unlike in Experiments 1-2. We also discuss the points raised by the Reviewer in the General Discussion (see p. 47).

Very minor

Reviewer 3 Comment #2:

- Grabenhorst et al (2019) might be a good recent reference for discussing observation learning in animals.

Authors: We have added this reference to the introduction as an illustration of the observational learning of preferences in animals (see p. 4).

Reviewer 3 Comment #3:

- In Exp1-2, in the instruction condition, did participants see a picture of the two logos?

Authors: We should clarify that the cookies in Experiments 1-2 differed purely in terms of their names (i.e., there were no brand logos). In other words, the cookies were simply referred to by their names in both the videos and the instructions. This will hopefully be clear to readers in the revised version as a methods figure has been included in response to Reviewer 2 Comment #7 (Figure 1).

Reviewer 3 Comment #4:

- In Exp1-2, did the authors measure the evaluative ratings BEFORE the observations? This might be relevant because participants may have some a priori preference toward one of the cookies.

Authors: We did not collect evaluative ratings before the acquisition phase. However, we do not consider this to be a problem because (a) the two CS names were selected based on being pre-rated as highly neutral by a separate sample as part of an earlier project; and (b) we counterbalanced which CS was paired with which model reaction in order to prevent any remaining a priori preferences from influencing the main results.

Reviewer 3 Comment #5:

- For Exp1-2, although these results are straightforward, it would still be great to include a figure.

Authors: We have included figures that show the results in Experiment 1 (Figures 2A-2B) and Experiment 2 (Figures 3A-3B).

Reference:

Grabenhorst, F., Báez-Mendoza, R., Genest, W., Deco, G., & Schultz, W. (2019). Primate amygdala neurons simulate decision processes of social partners. Cell, 177(4), 986-998.

Olsson, A., Knapska, E., & Lindström, B. (2020). The neural and computational systems of social learning. Nature Reviews Neuroscience, 21(4), 197-212.

Zhang, L., & Gläscher, J. (2020). A brain network supporting social influences in human decision-making. Science advances, 6(34), eabb4159.

References

- De Houwer, J. (2018). Propositional models of evaluative conditioning. *Social Psychological Bulletin*, 13(3), e28046. <https://doi.org/10.5964/spb.v13i3.28046>
- De Houwer, J., Van Dessel, P., & Moran, T. (2021). Attitudes as propositional representations. *Trends in Cognitive Sciences*, 25(10), 870-882. <https://doi.org/10.1016/j.tics.2021.07.003>
- Haaker, J., Golkar, A., Selbing, I., & Olsson, A. (2017). Assessment of social transmission of threats in humans using observational fear conditioning. *Nature Protocols*, 12(7), 1378–1386. <https://doi.org/10.1038/nprot.2017.027>
- Kelley, H. H., & Michela, J. L. (1980). Attribution theory and research. *Annual Review of Psychology*, 31(1), 457–501. <https://doi.org/10.1146/annurev.ps.31.020180.002325>
- Lindström, B., Golkar, A., Jangard, S., Tobler, P. N., & Olsson, A. (2019). Social threat learning transfers to decision making in humans. *Proceedings of the National Academy of Sciences*, 116(10), 4732–4737. <https://doi.org/10.1073/pnas.1810180116>
- Olsson, A., & Phelps, E. A. (2004). Learned fear of “unseen” faces after Pavlovian, observational, and instructed fear. *Psychological Science*, 15(12), 822–828. <https://doi.org/10.1111/j.0956-7976.2004.00762.x>
- Rachman, S. (1977). The conditioning theory of fear acquisition: A critical examination. *Behaviour Research and Therapy*, 15(5), 375–387. [https://doi.org/10.1016/0005-7967\(77\)90041-9](https://doi.org/10.1016/0005-7967(77)90041-9)

Appendix B

Response to Decision Letter

Manuscript Title: Learning via instructions about observations: Exploring similarities and differences with learning via actual observations

Manuscript ID: RSOS-220059

Associate Editor Comments to Author (Dr Oliver Robinson)

Associate Editor Comment #1:

Thank you for a comprehensive revision. We are delighted to accept your paper pending the final small tweaks suggested by the reviewers.

Authors: We thank the Editor and Reviewers for their feedback and have made the requested changes to the paper. Please see below for our point-by-point response to the Reviewers' comments (page numbers refer to the version of the manuscript with tracked changes).

Reviewer Comments to Author

Reviewer: 3

Reviewer 3 Comment #1:

The authors did a great job in revising their manuscript. It is much clearer and the conceptual differences between their experiments were also made explicit and discussed in detail. Only a small thing: one of the references I mentioned in my previous comment (Zhang & Gläscher, 2020) was not cited in the revised manuscript. And I'd also encourage the authors to mention a very recent work on observational fear learning (Haaker et al., 2021). Otherwise, I have no other comments, and congrats on this work.

Reference:

*Haaker, J., Diaz-Mataix, L., Guillazo-Blanch, G., Stark, S. A., Kern, L., LeDoux, J. E., & Olsson, A. (2021). Observation of others' threat reactions recovers memories previously shaped by firsthand experiences. *Proceedings of the National Academy of Sciences*, 118(30).*

*Zhang, L., & Gläscher, J. (2020). A brain network supporting social influences in human decision-making. *Science advances*, 6(34), eabb4159.*

Authors: We thank the Reviewer for these kind words. Both of the papers mentioned by the Reviewer are now cited in the manuscript (see p. 3 and p. 46).

Reviewer: 2

Reviewer 2 Comment #1:

The revised manuscript by Kasran and colleagues includes more detailed theoretical discussions, and I found the authors' response to the comments to be addressing most of major points. However, I'd like to further clarify a few points.

I appreciate the authors' clarification on the differences between learning via instructions and learning via instructions about observations, where informational contents differ. Related to this point and the theoretical implications, the key prediction under the propositional perspective seems to be that, when the information contents are matched (or, more precisely, if the learned information is matched), there should be no difference in learning, and the potential source of learning difference

between observations and instructions about observations can be attributed to the difference in information. To this end, authors have taken measures to match informational contents (Exp 2 and Exp 4), and interpret the remaining differences in learning under this perspective.

However, I am still slightly confused about the rationale for repeating the same recorded videos. If participants interpret the repetition as “this model tasted the cookie once and I’m watching it three times,” participants’ own ratings of CS should not change as a function of the number of repetitions. However, if they interpret the repetition as “this model tasted the cookie three times and ALWAYS showed the same reaction,” the number of repetitions should strengthen the beliefs about the relationship between CS and UR. Even though consensus is beyond the scope of this paper (I am excited to hear that the authors are exploring the link to attribution in the lab!), I think it would be worth discussing whether participants’ beliefs about the stimuli in the observation condition would lead to different predictions under associative vs propositional theories. This point is less significant in Experiment 3 and 4, where participants were told that the model went through the pairings multiple times (and due to multiple types of video clips, it may have been less obvious that the same recordings were repeating), but I think it would be worth asking the participants about their interpretations of repeated recordings in future studies, in light of informational content and believability that authors discuss.

Authors: We thank the Reviewer for clarifying the earlier comment and agree that it may indeed be very relevant. If some participants believed that they had seen three different instances of the model tasting a cookie (rather than the same instance three times), this would constitute consistency information, and from a propositional perspective we would indeed expect the resulting observational learning effect to be larger (whereas from an associative perspective, only the number of repetitions and the strength of the resulting association would seem to be important). We have included a section in the general discussion where we mention this possibility as well as the recommendation to assess participants’ interpretation of the videos (see p. 47).

Reviewer 2 Comment #2:

Minor comment:

In Figures 1 and 4, it would be helpful to clarify in the caption that the videos that were shown to participants did not have the faces masked with labels. It may be more informative if these were replaced with illustrations, if there is a need to not reveal the stimuli in the paper.

Authors: We have clarified in both figure captions that the faces have been masked for the figure, but that this was not the case in the videos shown to participants.

Reviewer: 1

Reviewer 1 Comment #1:

In their revision, the authors have generally clarified the issues I was concerned with. The phenomenon is of interest, and the authors’ presentation is nuanced.

On the more general point: the authors have provided more information about the propositional perspective. Based on this information, I really wonder what the difference to model-based reinforcement learning is? In model-based RL, the model typically includes both information about the value and identity of the outcome, and an estimate of the probability that the outcome will transpire. As such, such a model would specify whether the CS “sometimes predict” the US, and the identity of the US. It would be helpful to acknowledge this similarity.

Although I appreciate that a full theoretical exegesis is beyond the scope of the present paper, I urge

the authors to, in future work, integrate their propositional account with modern, computational accounts of learning, rather than rely on verbal contrast against various (strawman, in my opinion) “associative” accounts of learning.

Authors: We agree with the Reviewer that model-based reinforcement learning exhibits strong similarities with the propositional perspective. We now acknowledge this similarity in the introduction (see p. 9 and 11).

Reviewer 1 Comment #2:

Minor: the random effects structure used in the linear mixed models (random intercept by subject) is known to be too liberal in many cases (see <https://www.ncbi.nlm.nih.gov/pmc/articles/PMC3881361/> & <https://www.ncbi.nlm.nih.gov/pmc/articles/PMC2657178/>). Given that the authors are analyzing repeated measures data, the correct (in terms of error control) random effects structure usually includes random slopes for all fixed effects (including interactions).

Authors: We thank the Reviewer for drawing our attention to this issue. We have estimated a linear mixed model with the maximal random effects structure supported by the design (as defined in the first reference), including a random intercept and random slopes for all within-participant effects (CS, block, and CS x block). Although the fit was singular and we therefore also estimated a model with a simplified random effects structure (see p. 32-33), the interpretation of the fixed effects was the same for both models. Therefore, in line with the Reviewer’s recommendation and with Singmann and Kellen (2019), we report the statistics for the (theoretically more justifiable) maximal model (see p. 34). We have also updated the OSF project to include the new analysis code and changed the OSF link in the paper to the link for the new registration (see p. 11, 51, and 58).

Reference:

Singmann, H., & Kellen, D. (2019). An introduction to mixed models for experimental psychology. In D. Spieler & E. Schumacher (Eds.), *New Methods in Cognitive Psychology* (pp. 4–31). Routledge. <https://doi.org/10.4324/9780429318405-2>